# Chromatin remodeller Fun30<sup>Fft3</sup> induces nucleosome disassembly to facilitate RNA polymerase II elongation

Junwoo Lee[1,*], Eun Shik Choi[1,*], Hogyu David Seo[1], Keunsoo Kang[2], Joshua M. Gilmore[3], Laurence Florens[3], Michael P. Washburn[3,4], Joonho Choe[1], Jerry L. Workman[3] & Daeyoup Lee[1]

Previous studies have revealed that nucleosomes impede elongation of RNA polymerase II (RNAPII). Recent observations suggest a role for ATP-dependent chromatin remodellers in modulating this process, but direct *in vivo* evidence for this is unknown. Here using fission yeast, we identify Fun30<sup>Fft3</sup> as a chromatin remodeller, which localizes at transcribing regions to promote RNAPII transcription. Fun30<sup>Fft3</sup> associates with RNAPII and collaborates with the histone chaperone, FACT, which facilitates RNAPII elongation through chromatin, to induce nucleosome disassembly at transcribing regions during RNAPII transcription. Mutants, resulting in reduced nucleosome-barrier, such as deletion mutants of histones H3/H4 themselves and the genes encoding components of histone deacetylase Clr6 complex II suppress the defects in growth and RNAPII occupancy of cells lacking Fun30<sup>Fft3</sup>. These data suggest that RNAPII utilizes the chromatin remodeller, Fun30<sup>Fft3</sup>, to overcome the nucleosome barrier to transcription elongation.

[1] Department of Biological Sciences, Korea Advanced Institute of Science and Technology, Daejeon 34141, South Korea. [2] Department of Microbiology, Dankook University, Cheonan, Chungnam 31116, South Korea. [3] Stowers Institute for Medical Research, Kansas City, Kansas City, Missouri 64110, USA. [4] Department of Anatomy and Cell Biology, University of Kansas Medical Center, 3901 Rainbow Boulevard, Kansas City, Kansas 66160, USA. * These authors contributed equally to this work. Correspondence and requests for materials should be addressed to D.L. (email: daeyoup@kaist.ac.kr).

Nucleosomes impose a strong barrier against RNAPII-mediated transcription elongation *in vitro*[1,2]. Thus, RNAPII must overcome the nucleosome barrier during elongation *in vivo*; however, the mechanism by which RNAPII transcribes through the nucleosomes positioned at transcribing regions of genes remains poorly understood. The best characterized factor which modulates the nucleosome barrier during RNAPII elongation is the histone chaperone, FACT (FAcilitates Chromatin Transcription). FACT is known to promote RNAPII transcription through chromatin templates *in vitro*[3] and co-localizes with elongating RNAPII *in vivo*[4,5]. FACT was initially shown to displace histone H2A–H2B dimers from nucleosomes during transcription[6], but a recent report proposed an alternative model in which FACT abrogates the nucleosome barrier by disrupting histone-DNA contacts without causing dimer removal[7]. In budding yeast, FACT mutants show a transcription-dependent loss of nucleosomes, suggesting that FACT also contributes to reassembling nucleosomes after the passage of RNAPII[6,8]. FACT mutants have been associated with the initiation of cryptic transcription from within transcribing regions[4,9], indicating that the FACT-mediated recovery of normal chromatin at transcribing regions helps ensure the precision of transcription. Thus, RNAPII elongation requires that nucleosome dynamics be under a sophisticated level of control, such that nucleosomes are disrupted ahead of the passage of RNAPII and then immediately restored thereafter. DNA replication-independent histone exchange is highly active at promoters but is prevented at coding regions by FACT, suggesting that FACT recycles the original histones to reassemble nucleosomes behind the elongating RNAPII[10–13]. The exact mechanisms through which FACT modulates the nucleosome barrier during the passage of RNAPII still remains to be determined, as does the potential involvement of other factors in regulating nucleosome dynamics during RNAPII elongation.

Histone acetylation also contributes to modulating the nucleosome barrier during RNAPII elongation. The acetylation of histones H3 and H4 ahead of an elongating RNAPII are thought to destabilize nucleosomes. Notably, however, this acetylation must be removed to restore the normal chromatin state at transcribing regions. In *Saccharomyces cerevisiae*, the histone deacetylase complex, Rpd3S, is localized to transcribing regions by Set2-mediated H3 K36 methylation and by elongating RNAPII, and is responsible for removing histone acetylation at transcribing regions[14–16]. In the absence of the Rpd3S complex, histone acetylation is not properly removed from the transcribed region, leading to the same sort of cryptic intragenic transcription observed in FACT mutants[10,16].

ATP-dependent chromatin remodellers can displace histone dimers from nucleosomes or evict whole histone octamers from chromatin[17]. Using these and other remodelling activities, chromatin remodellers play pivotal roles in regulating nucleosome dynamics at promoters. For example, the SWR1 and INO80 complexes regulate the exchange of the histone variant dimer, H2A.Z–H2B, with the canonical histone dimer, H2A–H2B, in nucleosomes positioned near promoters[18–20]. The SWI/SNF and RSC complexes, in contrast, induce the disassembly of nucleosomes at promoters to maintain nucleosome-depleted regions (NDRs)[21–24]. Given their ability to disrupt nucleosomes at promoters, chromatin remodellers have long been suspected to be involved in antagonizing the nucleosome barrier within transcribing regions during RNAPII elongation. Indeed, the RSC complex was shown to promote RNAPII transcription through a nucleosomal template *in vitro*[25], and it was shown to localize to coding sequences, where it appears to regulate the occupancy of RNAPII and

histones[26–28]. Finally, mouse Chd1 has been implicated in the ability of RNAPII to overcome the nucleosome barrier and escape the promoter[29]. Despite these tantalizing observations, however, we lack conclusive evidence demonstrating that chromatin remodellers contribute to overcoming the nucleosome barrier to enable RNAPII elongation *in vivo*.

In this study, we used fission yeast to explore the involvement of chromatin remodellers in regulating the nucleosome barrier at transcribing regions. We report that Fun30[Fft3] promotes RNAPII occupancy at coding regions in collaboration with the additional Fun30 paralogue, Fft2, and the FACT complex. Furthermore, we show that Fun30[Fft3] interacts and co-localizes with RNAPII and transcription elongation factors. Importantly, we demonstrate that Fun30[Fft3] cooperates with FACT to induce nucleosome disassembly at transcribing regions, which accounts for a large fraction of RNAPII-mediated nucleosome disassembly. The slow growth of cells with impaired Fun30[Fft3] can be rescued by mutations that reduce the nucleosome barrier at transcribing regions, such as mutations in the genes encoding components of Clr6 complex II (equivalent to the Rpd3S complex in budding yeast) and histones themselves. We also show that a histone gene knockdown rescues the defects of RNAPII occupancy at transcribing regions in cells lacking Fun30[Fft3]. Collectively, these results suggest that Fun30[Fft3] plays a major role in nucleosome disassembly during RNAPII elongation, which facilitates the efficient passage of RNAPII through chromatin templates during transcription elongation.

## Results

**Fun30s promote RNAPII occupancy at transcribing regions.** To gain a comprehensive understanding of the roles of ATP-dependent chromatin remodellers in the regulation of RNAPII transcription, we performed a genome-wide analysis of RNAPII distribution among cells harbouring mutations of fission yeast chromatin remodellers. RNAPII is differentially phosphorylated at its C-terminal repeat domain (CTD) during different stages of transcription: it is phosphorylated at Ser5 (CTD$_{S5P}$) during transcription initiation[30], but becomes phosphorylated at Ser2 (CTD$_{S2P}$) on the transition to elongation[31]. We focused on RNAPII with CTD phosphorylation of Ser2 (RNAPII-CTD$_{S2P}$), as this represents the actively elongating form of RNAPII. However, a recent mass-spectrometric analysis of budding yeast revealed that CTD phosphorylation occurs in less than half of all CTD repeats, and that Ser2 phosphorylation is observed in fewer than 10% of all CTDs[32,33]. Thus, to assess the distribution of RNAPII regardless of CTD phosphorylation, we analysed RNAPII with non-phosphorylated CTDs (RNAPII-CTD$_{un}$) in parallel with RNAPII-CTD$_{S2P}$.

We profiled RNAPII occupancy changes in mutants defective in major chromatin-remodelling complexes, such as CHD (*hrp1Δ* and *hrp3Δ*), INO80 (*ies6Δ*), RSC (*snf21-36*), SWI/SNF (*snf22Δ*), and SWR1 (*swr1Δ*). We also analysed a mutant defective in the most recently identified chromatin remodeller Fun30 which has three paralogues (Fft1, Fft2 and Fft3) in fission yeast. Consistent with a previous study, we found that *fft3Δ* cells show a cold-sensitive growth defect[34] that becomes more severe when combined with *fft2Δ* but not with *fft1Δ* (Supplementary Fig. 1). In addition to cold-sensitivity, *fft3Δ* caused sensitivities to various stresses either alone or when combined with *fft2Δ* (Supplementary Fig. 1). We observed that *fft2Δ* did not cause any noticeable phenotype on its own and that *fft1Δ* did not cause any growth defect under normal and stress conditions even when combined with mutations in other Fun30 paralogues (Supplementary Fig. 1).

This suggests that among the Fun30 paralogues in fission yeast Fun30$^{Fft3}$ performs a major function in a partially redundant manner with Fun30$^{Fft2}$. Thus, we used *fft2Δ fft3Δ* cells as fission yeast cells lacking Fun30 function. Since we were interested in whether certain chromatin remodellers may contribute to regulating nucleosome dynamics at transcribing regions during RNAPII elongation as the histone-chaperone FACT complex does, we also included a FACT mutant (*spt16-18*) in the analysis. Metagene analysis was performed on RNAPII-CTD$_{S2P}$ and RNAPII-CTD$_{un}$ occupancy profiles obtained by chromatin immunoprecipitation combined with sequencing (ChIP-Seq). Our results revealed that RNAPII occupancy was severely reduced at genes with high RNAPII (RNAPII-CTD$_{S2P}$ and RNAPII-CTD$_{un}$) occupancy in *spt16-18*,

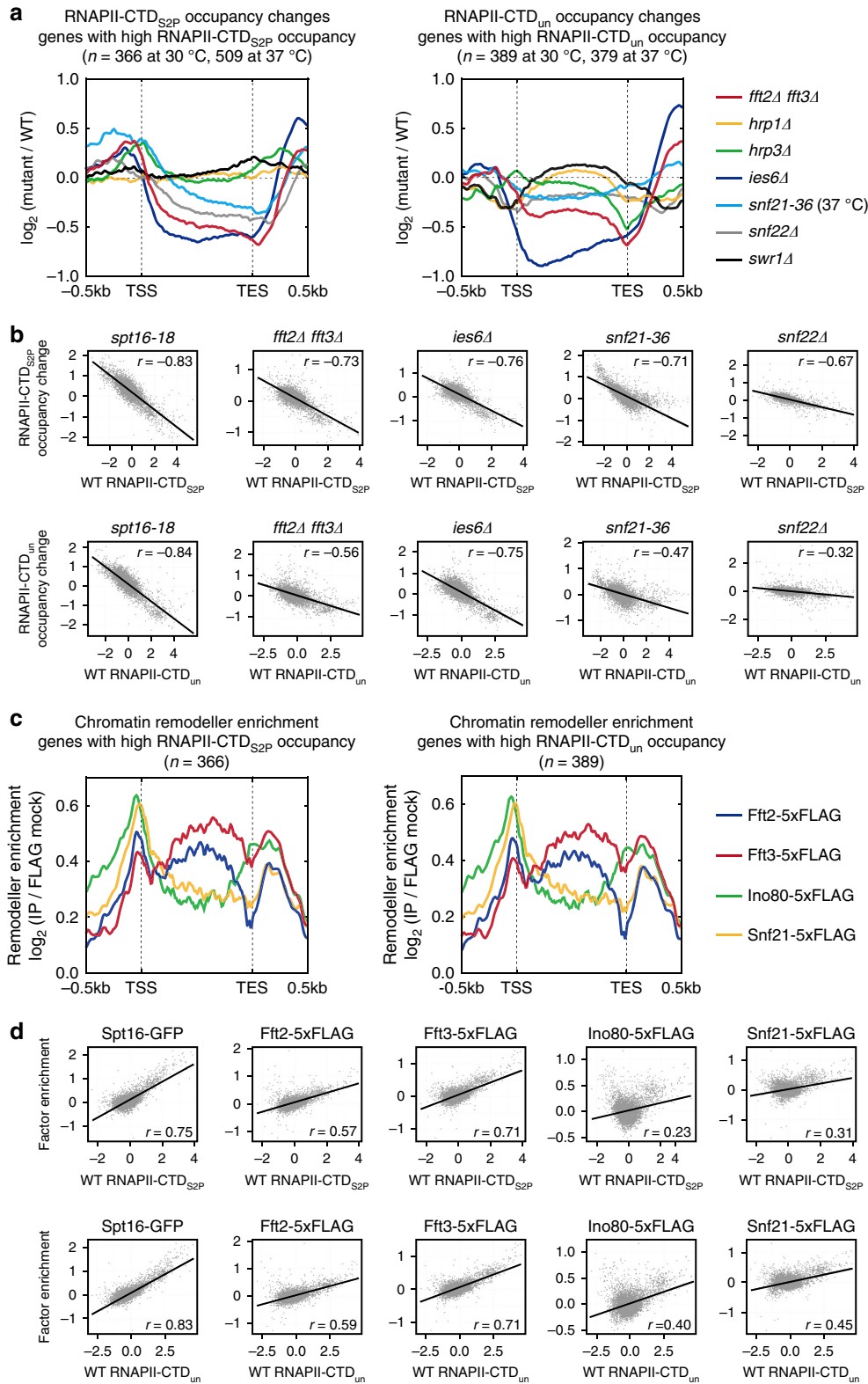

*fft2Δ fft3Δ*, *ies6Δ*, *snf21-36* and *snf22Δ* cells, but not in *hrp1Δ*, *hrp3Δ*, or *swr1Δ* cells (Fig. 1a and Supplementary Fig. 2a). Scatterplot analysis was used to calculate the correlation between RNAPII occupancy in wild-type cells and the changes in RNAPII occupancy among the mutant strains (Fig. 1b). We considered the absolute values of correlation coefficients greater than 0.4 ($r \geq 0.4$) as a meaningful correlation[35]. This analysis revealed that mutants of chromatin remodellers (for example, *ies6Δ*, *fft2Δ fft3Δ*, and *snf21-36* strains) and the FACT mutant (*spt16-18*) all exhibited transcription-dependent decreases in RNAPII occupancy (for both RNAPII-CTD$_{S2P}$ and RNAPII-CTD$_{un}$). Thus, it appears that these chromatin remodellers function during active transcription.

To determine the localizations of Ino80, Fun30$^{Fft2}$, Fun30$^{Fft3}$ and Snf21 across RNAPII-transcribed genes and the transcription-dependency of these localizations, we profiled the gene occupancies of Fun30$^{Fft2}$, Fun30$^{Fft3}$, Ino80, Snf21 and the essential subunit of the FACT complex, Spt16 (Fig. 1c and Supplementary Fig. 2b). Consistent with their transcription-dependent promotion of RNAPII occupancy, the chromatin remodellers and FACT showed significant binding to genes with high levels of RNAPII (RNAPII-CTD$_{S2P}$ and RNAPII-CTD$_{un}$) occupancy (Fig. 1c and Supplementary Fig. 2b). In contrast to Spt16, which is exclusively localized to transcribing regions, Ino80 and Snf21 were found to be preferentially localized to promoters and terminators as observed in budding yeast[27,36]. Notably, unlike Ino80 and Snf21, Fun30$^{Fft3}$ showed a higher enrichment over transcribing regions than at promoters or terminators (Fig. 1c and Supplementary Fig. 2b). Fun30$^{Fft2}$ was significantly enriched over transcribing regions (Fig. 1c and Supplementary Fig. 2b), but also localized to promoters and terminators (Fig. 1c and Supplementary Fig. 2b). Consistent with a recent report, we also detected a significant enrichment of Fun30$^{Fft2}$ at retrotransposon-flanking long terminal repeat (LTR) elements[34] (Supplementary Fig. 2c). This indicates that there is at some degree of specificity in the localizations and functions of Fun30$^{Fft2}$ and Fun30$^{Fft3}$. Scatterplot analysis was performed to calculate the correlation between the occupancy of RNAPII and those of FACT or the chromatin remodellers (Fig. 1d). As expected, FACT occupancy demonstrated the strongest correlation with RNAPII occupancy at transcribing regions, with Pearson correlation coefficients (r) = 0.75 and 0.83 in RNAPII-CTD$_{S2P}$ and RNAPII-CTD$_{un}$, respectively. Among the chromatin remodellers, only the occupancy of Fun30$^{Fft3}$ showed a strong correlation with that of RNAPII, with Pearson correlation coefficients ($r$ = 0.71 in both

RNAPII-CTD$_{S2P}$ and RNAPII-CTD$_{un}$) that were only slightly weaker than those obtained for FACT (Fig. 1d). The correlation of Fun30$^{Fft2}$ occupancy with RNAPII occupancy ($r$ = 0.57 and 0.59 in RNAPII-CTD$_{S2P}$ and RNAPII-CTD$_{un}$, respectively) was weaker than that of Fun30$^{Fft3}$ but stronger than those of Ino80 and Snf21. Our results indicate that there are strong similarities in the localizations and functions of FACT and the Fun30 paralogues in fission yeast. We also found that direct comparison of RNAPII occupancy changes of *fft2Δ fft3Δ* cells positively correlate with those of *spt16-18* cells ($r$ = 0.42 and 0.50 for RNAPII-CTD$_{S2P}$ and RNAPII-CTD$_{un}$; Supplementary Fig. 2d). Furthermore, the localizations of Fun30$^{Fft2}$ and Fun30$^{Fft3}$ at transcribing regions with that of Spt16 showed that Fun30 paralogues localize to transcribing regions in a manner similar to FACT ($r$ = 0.60 and 0.76 for Fun30$^{Fft2}$ and Fun30$^{Fft3}$; Supplementary Fig. 2e), suggesting that Fun30 paralogues may, similar to FACT, function at transcribing regions to promote RNAPII occupancy.

**Fun30$^{Fft3}$ and FACT share a common function in transcription.** Since Fun30$^{Fft3}$ and FACT exhibited similar distributions and transcription-dependent functions in promoting RNAPII occupancy, we checked whether there was any genetic interaction between Fun30$^{Fft3}$ and FACT. We combined *fft3Δ* and a temperature-sensitive (*ts*) mutation of *spt16*$^{+}$ (*spt16-1* or *spt16-2*), and found that *fft3Δ* causes synthetic growth defects when combined with *spt16-1* or *spt16-2* (Fig. 2a). Such synthetic growth defects were not observed when *fft3Δ* was combined with mutations in other transcription elongation-related factors such as Paf1 and Spt6 (*paf1Δ* and *spt6-1*). This revealed an essential function that is shared by Fun30$^{Fft3}$ and FACT but not apparently by Fun30$^{Fft3}$ and other transcription elongation-related factors. The synthetic growth defects of *fft3Δ spt16-1* and *fft3Δ spt16-2* cells were observed at a temperature (30 °C) that only partially disrupted the function of FACT. Thus, we could not determine whether or not the shared function of Fun30$^{Fft3}$ and FACT resides in the same pathway by the genetic analysis alone. To address this question, we compared the RNAPII occupancy changes in *fft3Δ spt16-1* cells with those in *fft3Δ* cells and in *spt16-1* cells grown at their restrictive temperature (37 °C). Scatterplot analyses revealed that *fft3Δ* did not have any additive effect on RNAPII occupancy when the function of Spt16 was severely impaired (Supplementary Fig. 3a,b), suggesting that Fun30$^{Fft3}$ and Spt16 may act in the same pathway to promote RNAPII occupancy.

**Figure 1 | The Fun30 family proteins promote RNAPII occupancy and localize to transcribing regions in correlation with RNAPII occupancy.** (**a**) Metagene profiles of RNAPII occupancy changes at genes with high RNAPII occupancy in representative chromatin remodeller mutants. All mutant and wild-type strains were grown at 30°C except the temperature-sensitive strain, *snf21-36*, which was temperature shifted from 25 °C to 37 °C and, along with its wild-type control, grown at 37°C for 2 hours. TSS, transcription start site; TES, transcription termination site. The RNAPII-CTD$_{S2P}$ and RNAPII-CTD$_{un}$ occupancy profiles in wild-type cells grown at 30°C or 37°C were individually subject to k-means clustering ($k$ = 3) to determine genes with high RNAPII-CTD$_{S2P}$ occupancy and those with high RNAPII-CTD$_{un}$ occupancy. (**b**) Scatterplots showing the correlation between the occupancy of RNAPII-CTD$_{S2P}$ and RNAPII-CTD$_{un}$ in wild-type cells (log$_2$(IP/INPUT)) and the occupancy changes of RNAPII-CTD$_{S2P}$ and RNAPII-CTD$_{un}$ in chromatin remodeller mutants (log$_2$(mutant/WT)). Scatterplot analysis of results obtained using a FACT mutant (*spt16-18*) was performed as a positive control. RNAPII occupancies (IP/INPUT) at ORFs of 5,150 protein-coding genes were calculated by using merged reads obtained from biological duplicates. Correlation coefficients (r) were calculated by the Pearson method. The r values were −0.83, −0.73, −0.76, −0.71 and −0.67 for transcription-dependent RNAPII-CTD$_{S2P}$ occupancy change, and −0.84, −0.56, −0.75, −0.47 and −0.32 for transcription-dependent RNAPII-CTD$_{un}$ occupancy change of *spt16-18*, *fft2Δfft3Δ*, *ies6Δ*, *snf21-36* and *snf22Δ* cells. (**c**) Metagene profiles for enrichment of Fun30$^{Fft2}$, Fun30$^{Fft3}$, Ino80 and Snf21 at transcribing regions of genes with high RNAPII occupancy. The genes with high RNAPII (RNAPII-CTD$_{S2P}$ and RNAPII-CTD$_{un}$) occupancy are the same as presented in (A). (**d**) Scatterplots showing the correlation between the occupancy of RNAPII (RNAPII-CTD$_{S2P}$ and RNAPII-CTD$_{un}$; log$_2$(IP/INPUT)) and the enrichment of chromatin remodellers at transcribing regions of wild-type cells (log$_2$(IP/mock)). The enrichments of chromatin remodellers (IP/INPUT) at ORFs of 5,150 protein-coding genes were calculated by using merged reads obtained from biological duplicates. Correlation coefficients (r) were calculated by the Pearson method. The r values were 0.75, 0.57, 0.71, 0.23 and 0.31 for RNAPII-CTD$_{S2P}$-dependent binding, and 0.83, 0.59, 0.71, 0.40 and 0.45 for RNAPII-CTD$_{un}$-dependent binding of Spt16-GFP, Fft2-5xFLAG, Fft3-5xFLAG, Ino80-5xFLAG and Snf21-5xFLAG.

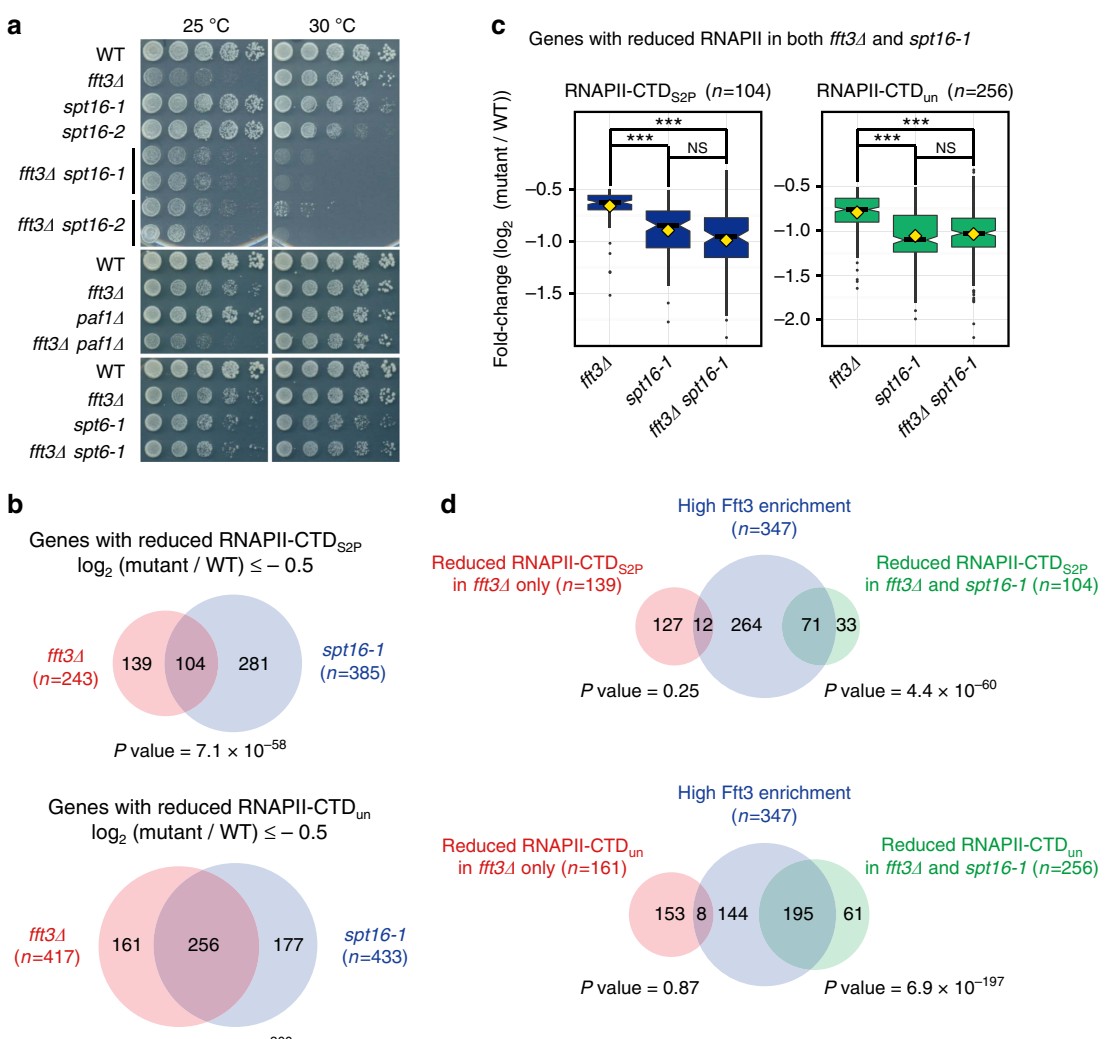

**Figure 2 | The role of Fun30$^{Fft3}$ in regulating RNAPII occupancy largely overlaps with that of Spt16.** (**a**) A spotting assay was used to assess the genetic interactions between Fun30$^{Fft3}$ and transcription elongation factors such as Spt16, Paf1 and Spt6. Serial dilutions (5-fold) of wild-type, *fft3Δ*, *spt16-ts* (*spt16-1* and *spt16-2*), *paf1Δ*, *spt6-1*, *fft3Δ spt16-ts* (*spt16-1* and *spt16-2*), *fft3Δ paf1Δ* and *fft3Δ spt6-1* cells were spotted on rich medium and grown at the permissive (25 °C) and semi-permissive (30 °C) temperatures of *spt16-ts* cells. Cells grown at 25°C were incubated for an additional day to allow full growth. (**b**) Venn diagram depicting the overlap of genes showing decreased RNAPII occupancy at transcribing regions (log$_2$ fold-change ≤ − 0.5) in *fft3Δ* versus *spt16-1* cells. *P* values for the significance of list overlap were calculated according to hypergeometric distribution. (**c**) Boxplots of RNAPII occupancy changes at transcribing regions of *fft3Δ*, *spt16-1*, and *fft3Δ spt16-1* cells for the overlapping genes identified in (b). The upper and lower whiskers extend from the upper and lower hinges to the highest and the lowest value that are within 1.5 × interquartile range (IQR), and the dots represent outliers. *P* values were calculated according to Tukey's multiple comparison test. NS, *P* ≥ 0.1; ***$P$ ≤ 0.001. (**d**) Venn diagram depicting the overlap of genes which show high Fun30$^{Fft3}$ occupancies at transcribing regions and those which show decreased RNAPII occupancies (log$_2$ fold-change ≤ − 0.5) in both *fft3Δ* cells and *spt16-1* cells (right) or only in *fft3Δ* cells (left). *P* values for the significance of list overlap were calculated according to hypergeometric distribution. The Fun30$^{Fft3}$ occupancy profiles in wild-type cells grown at 30 °C were subject to k-means clustering ($k = 3$) to determine genes with high Fun30$^{Fft3}$ occupancy.

To further explore the functional overlap between Fun30$^{Fft3}$ and FACT, we compared genes that exhibited significantly reduced RNAPII occupancies in *fft3Δ* versus *spt16-1* cells (Fig. 2b). Venn diagram analysis demonstrated that there was a significant overlap of genes whose RNAPII (RNAPII-CTD$_{S2P}$ and RNAPII-CTD$_{un}$) occupancies were reduced in *fft3Δ* cells (log$_2$ fold-change ≤ − 0.5) versus *spt16-1* cells (log$_2$ fold-change ≤ − 0.5). Analysis of the overlapping genes by boxplots showed that the reduction of RNAPII occupancy in *spt16-1* cells was more severe than that in *fft3Δ* cells, but was similar to that in *fft3Δ spt16-1* cells (Fig. 2c). A similar finding was obtained using a genome-browser view (Supplementary Fig. 3c). We also found that genes at which Fun30$^{Fft3}$ was highly enriched significantly overlapped with genes whose RNAPII occupancies were reduced in both *fft3Δ* cells and *spt16-1* cells, but not with those whose RNAPII occupancies were reduced only in *fft3Δ* cells (Fig. 2d). Furthermore, the levels of decreased RNAPII occupancy in *fft3Δ* cells at the genes whose RNAPII occupancies were reduced only in *fft3Δ* cells did not obey the trend of transcription-dependent decrease in RNAPII occupancy observed in *fft3Δ* cells (Supplementary Fig. 3d). These findings suggest that direct regulation of RNAPII occupancy by Fun30$^{Fft3}$ during transcription is mediated in a FACT-dependent manner.

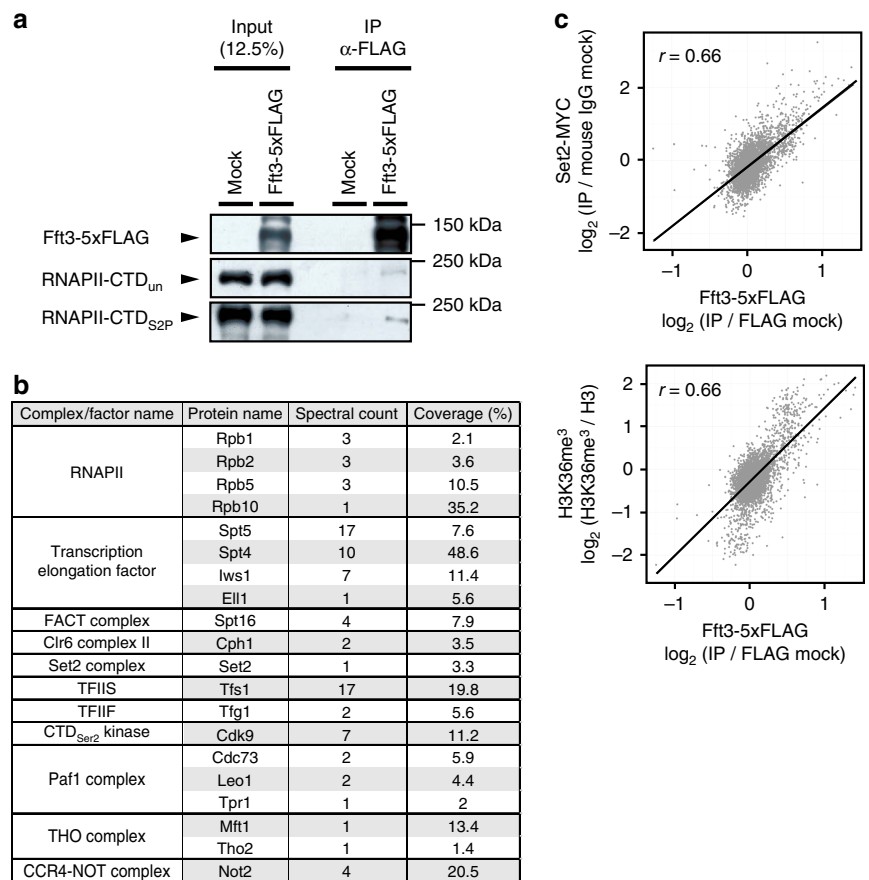

**Figure 3 | Fun30$^{Fft3}$ interacts and co-localizes with RNAPII and transcription elongation-related factors.** (**a**) FLAG-tagged Fft3 was immunoprecipitated using anti-FLAG antibodies, and the immunoprecipitates were subjected to western blot analysis for the presence of RNAPII-CTD$_{un}$ or RNAPII-CTD$_{S2P}$. 'Mock' presents the results of a parallel Western blot analysis run using the untagged wild-type control strain. 'Input' represents 12.5% of the total lysate before immunoprecipitation. (**b**) Table showing subunits of RNAPII and transcription elongation-related proteins identified from MudPIT (Multi-Dimensional Protein Identification Technology) analysis following tandem affinity purification (TAP) of Fft3 complexes. (**c**) Scatterplot analyses were used to correlate the enrichments of Fft3 versus Set2 or H3K36me$^3$ at transcribing regions. The ChIP enrichments (IP/INPUT) at ORFs of 5,150 protein-coding genes were calculated by using merged reads obtained from biological duplicates. Correlation coefficients were calculated according to the Pearson method. The correlation coefficient values were both 0.66 for the comparison of Fft3-5xFLAG enrichment with Set2-MYC and H3K36me$^3$ enrichments.

**Fun30$^{Fft3}$ associates with transcription elongation factors.** The correlation between RNAPII occupancy and the localization or function of Fun30$^{Fft3}$ at transcribing regions suggests that Fun30$^{Fft3}$ may interact with RNAPII. To test this possibility, we performed co-immunoprecipitation assays using lysates of fission yeast cells expressing untagged or FLAG-tagged Fun30$^{Fft3}$. Indeed, Fun30$^{Fft3}$ co-immunoprecipitated both RNAPII-CTD$_{un}$ and RNAPII-CTD$_{S2P}$ (Fig. 3a; uncropped immunoblots for the Fig. 3a were described in Supplementary Fig. 8a), revealing that this Fun30 family member interacts with RNAPII in a CTD-phosphorylation-independent manner. In addition, MudPIT (Multi-Dimensional Protein Identification Technology) analysis of Fun30$^{Fft3}$ complexes purified using a TAP (Tandem Affinity Purification) method demonstrated that Fun30$^{Fft3}$ interacted not only with RNAPII subunits but also with various transcription elongation-related proteins, including Spt16 (Fig. 3b and Supplementary Table 1). Consistent with the physical interaction data, we observed that the occupancy of Fun30$^{Fft3}$ at transcribing regions strongly correlated with the occupancies of Set2 and the levels of Set2-mediated histone H3K36 trimethylation (H3K36me$^3$)[37] (Fig. 3c). Altogether, these results implicate that Fun30$^{Fft3}$ may regulate RNAPII transcription by interacting with both RNAPII and transcription-elongation factors.

**Fun30$^{Fft3}$ helps transcription-coupled nucleosome disassembly.** Since chromatin remodellers can induce nucleosome disassembly at promoters to reduce the nucleosome barrier, we speculated that Fun30$^{Fft3}$ localized to transcribing regions might perform this function during RNAPII elongation. To test this possibility, we first examined how nucleosome occupancy is regulated by Fun30$^{Fft3}$ at transcribing regions during transcription. Comparison of Fun30$^{Fft3}$-mediated nucleosome loss (histone H3 occupancy changes of fft3Δ cells relative to wild-type) at transcribing regions with RNAPII occupancies revealed that histone H3 occupancy is increased at a number of transcribing regions in fft3Δ cells especially among highly transcribed genes (Fig. 4a). However, the increased histone H3 occupancies in fft3Δ cells did not correlate with RNAPII occupancies ($r = 0.20$ and 0.33 for RNAPII-CTD$_{S2P}$ and RNAPII-CTD$_{un}$; Fig. 4a). The increase in histone H3 occupancies at transcribing regions caused by fft3Δ was more significant in cells with intact FACT than in cells with defective FACT, indicating that Fun30$^{Fft3}$ regulates nucleosome occupancy at transcribing regions in a FACT-dependent manner (Compare Fig. 4a with Supplementary Fig. 4a). To gain more insights into the role of Fun30$^{Fft3}$ in regulating nucleosome occupancy during transcription, we investigated whether Fun30$^{Fft3}$-mediated nucleosome loss at transcribing regions is

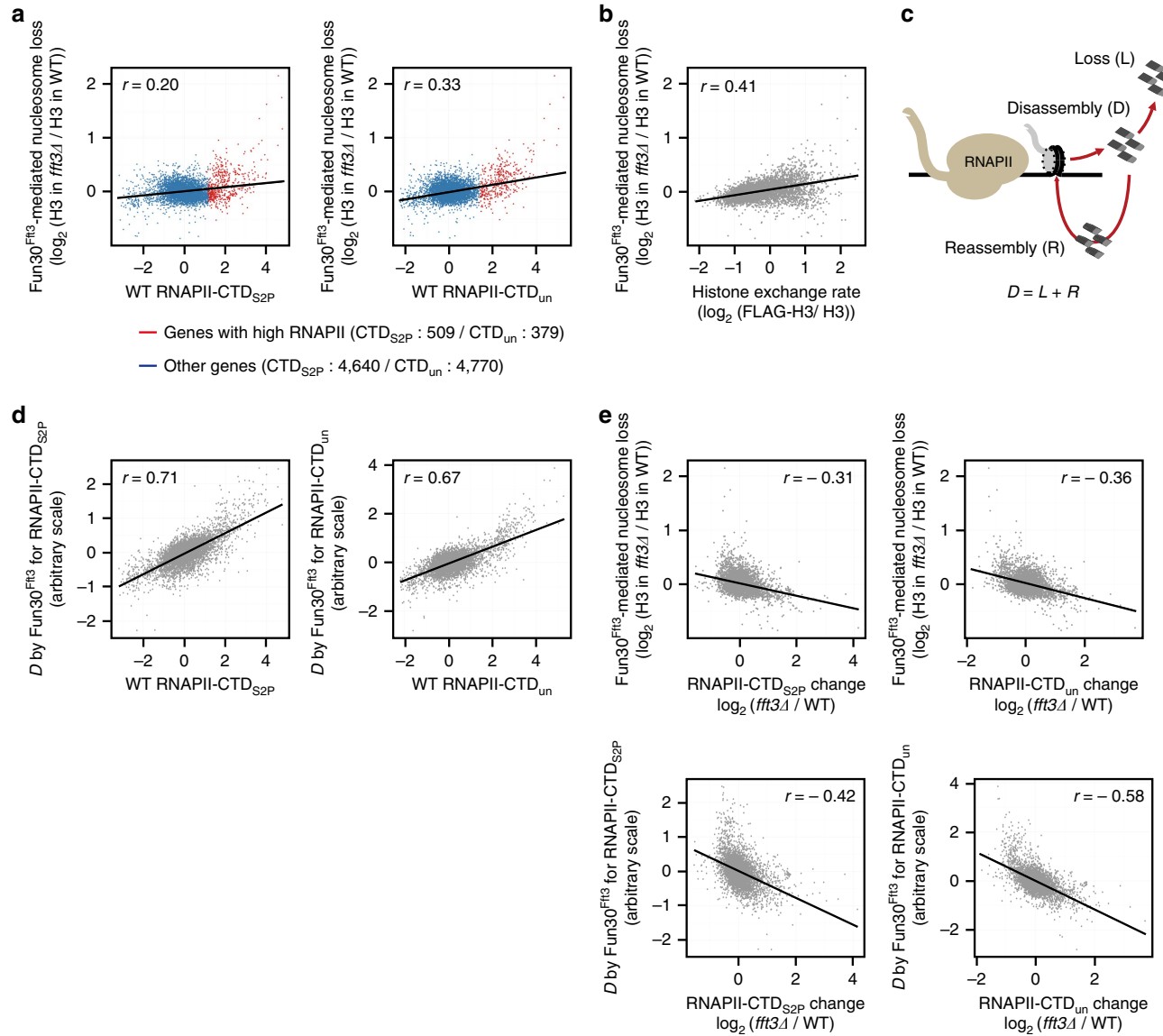

**Figure 4 | Fun30^Fft3 induces nucleosome disassembly at transcribing regions during RNAPII transcription. (a)** Scatterplots showing the correlation between the RNAPII occupancies in wild-type cells and the levels of Fun30^Fft3-mediated nucleosome loss at transcribing regions of genes with high RNAPII occupancies in wild-type cells (red; $n = 539$ and 379 for RNAPII-CTD$_{S2P}$ and RNAPII-CTD$_{un}$) or at transcribing regions of other genes (blue; $n = 4,640$ and 4,770 for RNAPII-CTD$_{S2P}$ and RNAPII-CTD$_{un}$). The numbers of genes with high levels of RNAPII occupancies were determined as described in Fig. 1a. Data were obtained from biological duplicates. The correlation coefficients ($r$) of all the scatterplots were calculated by the Pearson method. The r values were 0.20 and 0.33 for the comparison of Fun30^Fft3-mediated nucleosome loss with RNAPII-CTD$_{S2P}$ and RNAPII-CTD$_{un}$. **(b)** Scatterplot showing the correlation between the rate of histone H3 exchange in wild-type cells and the histone H3 occupancy changes at transcribing regions of $fft3\varDelta$ cells. Data were obtained from the 5,149 protein-coding genes of duplicate samples. Correlation coefficient value 0.41 was calculated by the Pearson method. **(c)** Schematic model showing the relation among the RNAPII-mediated nucleosome disassembly ($D$), the nucleosome loss ($L$) and the nucleosome reassembly ($R$). **(d)** Scatterplots showing the correlation between the RNAPII occupancies in wild-type cells and the estimated levels of Fun30^Fft3-mediated nucleosome disassembly at transcribing regions. Data were obtained from the 5,149 protein-coding genes of duplicate samples. Correlation coefficients ($r$) were calculated by the Pearson method. The r values were 0.71 and 0.67 for Fun30^Fft3-mediated nucleosome disassembly coupled with RNAPII-CTD$_{S2P}$ and RNAPII-CTD$_{un}$. For the method of estimation, see Methods section. **(e)** Scatterplots showing the correlation between the RNAPII occupancy changes in $fft3\varDelta$ cells and the levels of Fun30^Fft3-mediated nucleosome loss (top) or the estimated levels of Fun30^Fft3-mediated nucleosome disassembly at transcribing regions (bottom). Data were obtained from the 5,149 protein-coding genes of duplicate samples. Correlation coefficients ($r$) were calculated by the Pearson method. The $r$ values were $-0.31$ and $-0.36$ for the comparison of Fun30^Fft3-mediated nucleosome loss with RNAPII-CTD$_{S2P}$ and RNAPII-CTD$_{un}$ occupancy changes, and $-0.42$ and $-0.58$ for the comparison of Fun30^Fft3-mediated nucleosome disassembly with RNAPII-CTD$_{S2P}$ and RNAPII-CTD$_{un}$ occupancy changes.

related to nucleosome loss, which occurs at transcribing regions during transcription. It is known that nucleosome loss during transcription stimulates incorporation of new histones, especially histones H3 and H4, and thus is associated with

DNA replication-independent exchange of histone H3 and H4 (refs 8,38). Scatterplot analysis revealed that the levels of histone H3 occupancy changes at transcribing regions of $fft3\varDelta$ cells positively correlate with the rates of histone H3 exchange in

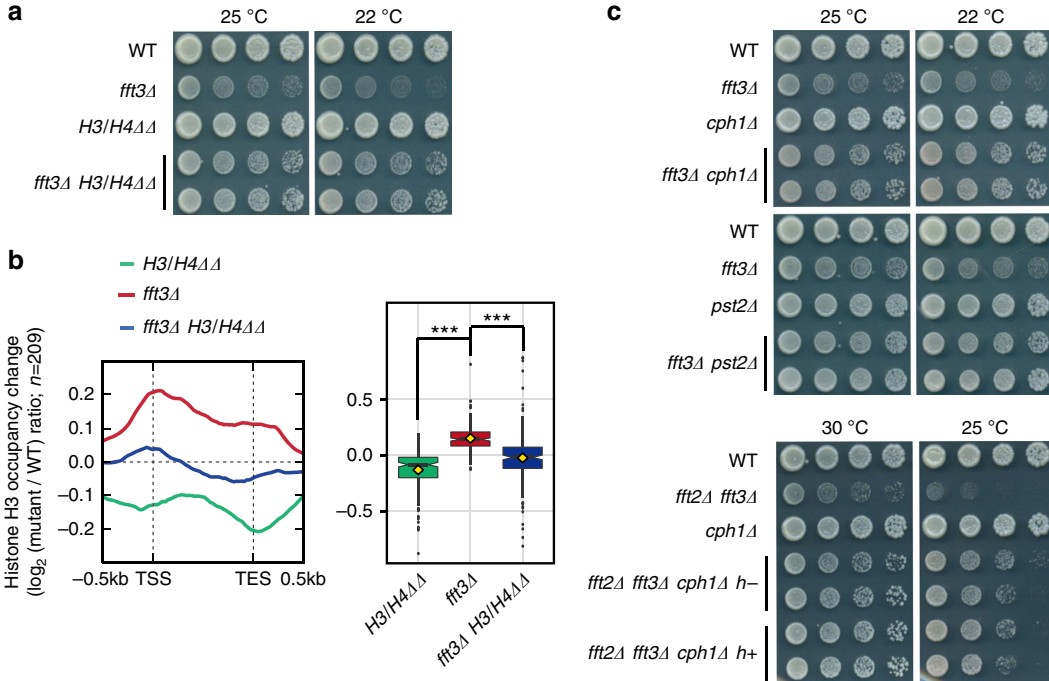

**Figure 5 | Reduction of the nucleosome barrier suppresses the growth defect of *fft3Δ* cells. (a)** Spotting assays were used to monitor the growth profiles of wild-type, *fft3Δ*, *H3/H4ΔΔ*, and *fft3Δ H3/H4ΔΔ* strains. Serial dilutions (fivefold) of cells were spotted and grown at 25°C or 22°C (with an additional day of incubation to allow full growth). **(b)** Metagene profile and boxplot showing the changes of histone H3 occupancy in *H3/H4ΔΔ* (green), *fft3Δ* (red), and *fft3Δ H3/H4ΔΔ* (blue) cells, with respect to genes that showed increased histone H3 occupancy in *fft3Δ* cells at 22°C. Target genes were grouped by k-means clustering (*k* = 5), based on their change of histone H3 occupancy in *fft3Δ* cells. The upper and lower whiskers extend from the upper and lower hinges to the highest and the lowest value that are within 1.5 × inter-quartile range (IQR), and the dots represent outliers. *P* values were calculated according to Tukey's multiple comparison test. NS, $P \geq 0.1$; ***$P \leq 0.001$. **(c)** Spotting assays were used to monitor the growth profiles of the wild-type, *fft3Δ*, *cph1Δ*, *pst2Δ*, *fft3Δ cph1Δ*, and *fft3Δ pst2Δ* strains at 25°C and 22°C (top), and those of the wild-type, *fft2Δ fft3Δ*, *cph1Δ*, and *fft2Δ fft3Δ cph1Δ* strains at 30°C and 25°C (bottom). The *fft2Δ fft3Δ cph1Δ* cells grown at 25°C and the *fft3Δ cph1Δ* and *fft3Δ pst2Δ* cells grown at 22°C were incubated for one more day than their higher-temperature-grown counterparts to allow comparable growth.

wild-type cells[39] (*r* = 0.41; Fig. 4b), indicating that Fun30^Fft3-mediated nucleosome loss at transcribing regions is related to histone exchange and, by extension, to nucleosome loss at transcribing regions during transcription.

The level of transcription-coupled nucleosome disassembly at transcribing regions should be proportional to the level of transcription (Fig. 4c). However, the actual level of nucleosome loss at transcribing regions by RNAPII transcription may not be strictly proportional to the level of transcription since nucleosomes that were transiently disassembled by RNAPII transcription can be efficiently reassembled during transcription and only some of the disassembled nucleosomes that failed to be reassembled may be eventually lost. Histone chaperones such as FACT and Spt6 play essential roles in nucleosome reassembly during RNAPII transcription[9,38,40] (Supplementary Fig. 4b). Thus, even when nucleosome disassembly by Fun30^Fft3 is coupled to transcription, nucleosome loss at transcribing regions induced by Fun30^Fft3 may not correlate with RNAPII occupancies. We sought to explore this possibility further by investigating whether transcription-coupled nucleosome disassembly at transcribing regions by Fun30^Fft3 can result in the observed histone H3 occupancy changes in *fft3Δ* cells using a simple mathematical model that fits the experimental data with minimal assumptions (see Methods section for more detail). In this model, we hypothesized that the level of RNAPII-mediated nucleosome disassembly at transcribing regions, denoted by *D* (steady state level of nucleosomes that are disassembled by RNAPII transcription at transcribing regions relative to total number of nucleosomes at transcribing

regions), is equivalent to the sum of the level of nucleosome loss during transcription, denoted by *L* (steady state level of nucleosomes that are disassembled and not immediately reassembled during RNAPII transcription at transcribing regions relative to total number of nucleosomes at transcribing regions), and the level of nucleosome reassembly during transcription, denoted by *R* (steady state level of nucleosomes that are transiently disassembled but immediately reassembled during transcription at transcribing regions relative to total number of nucleosomes at transcribing regions). By fitting the available experimental data to the model, we could estimate the relative contributions of Fun30^Fft3 in RNAPII-mediated nucleosome disassembly at transcribing regions to be 84% and 54% for RNAPII-CTD_{S2P} and RNAPII-CTD_{un}, respectively (see Methods section for more detail). The levels of Fun30^Fft3-mediated nucleosome disassembly at transcribing regions estimated in this way strongly correlated with RNAPII occupancies (*r* = 0.71 and 0.67 for RNAPII-CTD_{S2P} and RNAPII-CTD_{un}; Fig. 4d). This analysis validated the earlier hypothesis that the histone H3 occupancy changes in *fft3Δ* cells, which is seemingly unrelated to RNAPII transcription (Fig. 4a), can result from a transcription-coupled defect in RNAPII-mediated nucleosome disassembly (Fig. 4d). Furthermore, we found that RNAPII occupancy changes in *fft3Δ* cells correlate with the estimated levels of Fun30^Fft3–mediated nucleosome disassembly at transcribing regions (*r* = − 0.42 and − 0.58 for RNAPII-CTD_{S2P} and RNAPII-CTD_{un}) rather than the levels of Fun30^Fft3–mediated nucleosome loss at transcribing regions (*r* = − 0.31 and − 0.36 for RNAPII-CTD_{S2P} for RNAPII-CTD_{un}; Fig. 4e).

Together, these analyses suggest that Fun30$^{Fft3}$ plays a major role in RNAPII-mediated nucleosome disassembly at transcribing regions and this role is associated with promotion of RNAPII occupancy by Fun30$^{Fft3}$.

**Fun30$^{Fft3}$ reduces nucleosome barrier to RNAPII elongation.** To more directly determine whether the cell growth defects and reduced RNAPII occupancies associated with the loss of Fun30 paralogues were caused by an increase in the nucleosome barrier

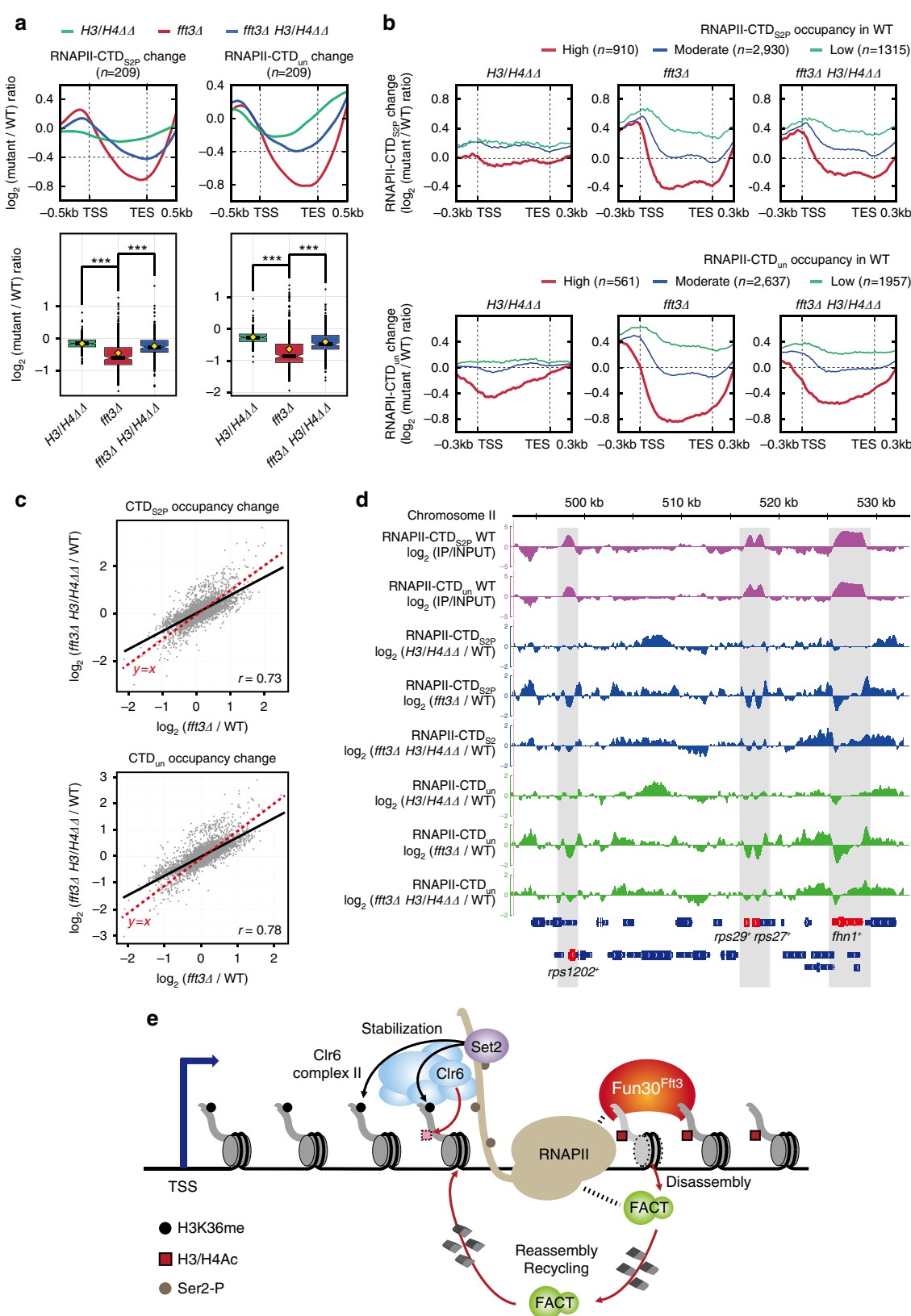

at transcribing regions, we tested whether mutations capable of reducing this barrier could suppress the phenotype of *fft3Δ* cells. To reduce the nucleosome barrier, we first introduced copy-number mutations that reduced the expression levels of the histone genes. Simultaneous deletion of two out of three copies of the genes encoding histone H3 and H4 (*H3/H4ΔΔ*) significantly reduced the amounts of soluble and chromatin-bound histone H3 and H4 (Supplementary Fig. 5a,b; primers used for Supplementary Fig. 5b were described in Supplementary Table 3; uncropped immunoblots for Supplementary Fig. 5a were described in Supplementary Fig. 8b). The *H3/H4ΔΔ* mutation suppressed the cold-sensitive growth phenotype observed in *fft3Δ* cells (Fig. 5a), indicating that nucleosomes mediate this growth defect (Fig. 5b). We also introduced a mutation in the histone deacetylase, Clr6 complex II (equivalent to the Rpd3S complex in budding yeast), that was shown to destabilize chromatin at transcribing regions by causing defects in the removal of histone acetylation[41]. Subunits of Clr6 complex II are non-essential except the catalytic subunit Clr6. A subunit of Clr6 complex II, Cph1 is a fission yeast homologue of budding yeast Rco1, a PHD zinc-finger domain containing protein of Rpd3S complex[41]. We found that *cph1Δ* suppressed the growth defects of *fft2Δ fft3Δ* cells and the cold sensitivity of *fft3Δ* cells (Fig. 5c). Another subunit Pst2 is a fission yeast homologue of Sin3 proteins which are conserved from budding yeast to mammals. We found that *pst2Δ* also suppressed the cold-sensitive growth defect of *fft3Δ* cells (Fig. 5c). These results indicate that the action of Fun30 paralogues on cell growth can be bypassed by relieving the nucleosome barrier at transcribing regions.

Next, we asked whether an artificial reduction of the nucleosome barrier could recover the occupancy of RNAPII in *fft3Δ* cells. To determine this, we selected genes whose histone H3 occupancy was increased in *fft3Δ* cells, and compared their RNAPII occupancies in *fft3Δ H3/H4ΔΔ* cells with those in *fft3Δ* cells and *H3/H4ΔΔ* cells grown at 22 °C. Metagene and boxplot analyses showed that the *fft3Δ H3/H4ΔΔ* mutation caused a significantly smaller decrease in the occupancy of RNAPII at these genes compared to the *fft3Δ* mutation (Fig. 6a), indicating that the histone-copy-number mutations (*H3/H4ΔΔ*) recovered the RNAPII occupancy of genes with increased histone H3 occupancy in *fft3Δ* cells. Metagene analysis of the transcribing regions for all protein-coding genes revealed that the transcription-dependent decrease in the RNAPII occupancy of *fft3Δ H3/H4ΔΔ* cells fell between those of *H3/H4ΔΔ* and *fft3Δ* cells (Fig. 6b). Scatterplot analysis revealed that the transcription-dependent decrease of the RNAPII occupancy at transcribing

regions in *fft3Δ H3/H4ΔΔ* cells was relatively weaker than that in *fft3Δ* cells, but the levels of decrease in the RNAPII occupancy in *fft3Δ H3/H4ΔΔ* cells still showed a high correlation with those in *fft3Δ* (Supplementary Fig. 6 and Fig. 6c). This suggests that the *H3/H4ΔΔ* mutation causes an incomplete but global suppression of the RNAPII occupancy defect in *fft3Δ* cells, as confirmed by inspection of individual genes (Fig. 6d). Collectively, these results provide strong evidence that Fun30[Fft3] promotes RNAPII occupancy by reducing the nucleosome barrier at transcribing regions.

## Discussion

The nucleosome landscape of a typical eukaryotic gene includes a nucleosome-depleted region (NDR) or unstable nucleosomes at the promoter followed by well-positioned nucleosomes over the transcribing regions that are aligned with the transcription start site. Previously, we did not understand in detail how RNAPII overcomes these well-positioned nucleosomes at transcribing regions during transcription elongation. Here we show that Fun30[Fft3] interacts with RNAPII and facilitates disassembly of nucleosomes at transcribing regions in a transcription-dependent manner (Figs 3 and 4). Estimation of the levels of Fun30[Fft3]-mediated nucleosome disassembly at transcribing regions showed that Fun30[Fft3] accounts for a major portion of nucleosome disassembly at transcribing regions by RNAPII transcription. The role for a chromatin remodeller in nucleosome disassembly at transcribing regions was unprecedented and suggests a more elaborate model for the regulation of nucleosome dynamics during RNAPII elongation. In this model, RNAPII utilizes the chromatin remodeller Fun30[Fft3] as a means to disassemble nucleosomes and to transit through the positioned nucleosomes at transcribing regions (Fig. 6e). Our results further demonstrated that Fun30[Fft3] acts in a manner that largely depends on FACT (Figs 2 and 4), suggesting that the two cooperate and act in the same pathway to abrogate the nucleosome barrier ahead of an elongating RNAPII (Fig. 6e). The FACT complex is known to induce the disassembly of histone H2A-H2B dimers *in vitro*, and has also been shown to be required for the reassembly of nucleosomes at transcribing regions during RNAPII elongation *in vivo*[6,8]. We found that defects in FACT but not Fun30[Fft3] caused loss of nucleosomes at transcribing regions (Fig. 4 and Supplementary Fig. 4b), indicating that Fun30[Fft3] is not required for the reassembly of nucleosomes. Our analysis also showed that defective nucleosome disassembly at transcribing regions in *fft3Δ* cells correlates with reduced RNAPII occupancy in

**Figure 6 | Reduction of the nucleosome barrier suppresses the global RNAPII occupancy defect seen in *fft3Δ* cells.** (**a**) Metagene profile and boxplot showing the changes of the RNAPII occupancies in *H3/H4ΔΔ* (green), *fft3Δ* (red) and *fft3Δ H3/H4ΔΔ* (blue) cells, with respect to genes that showed increased histone H3 occupancy in *fft3Δ* cells at 22 °C, clustered as shown in Fig. 5b. The upper and lower whiskers extend from the upper and lower hinges to the highest and the lowest value that are within 1.5 × interquartile range (IQR), and the dots represent outliers. *P* values were calculated according to Tukey's multiple comparison test. NS, $P \geq 0.1$; ***$P \leq 0.001$. (**b**) Metagene profiles representing RNAPII occupancy changes in *H3/H4ΔΔ*, *fft3Δ*, and *fft3Δ H3/H4ΔΔ* cells. Genes were grouped by high, moderate, and low RNAPII occupancy using k-means clustering ($k = 3$). (**c**) Scatterplots showing the correlations of RNAPII occupancy changes in *fft3Δ* and *fft3Δ H3H4ΔΔ* cells. The RNAPII occupancy changes at ORFs of 5,147 protein-coding genes were calculated by using merged reads obtained from biological duplicates. Correlation coefficients (r) were calculated by the Pearson method. The r values were 0.73 and 0.78 for the comparison between RNAPII-CTD$_{S2P}$ and RNAPII-CTD$_{un}$ occupancy changes in *fft3Δ* and *fft3Δ H3H4ΔΔ* cells. (**d**) Genome-browser view of a representative chromosome region showing the profiles of RNAPII occupancy changes in *H3H4ΔΔ*, *fft3Δ*, and *fft3Δ H3H4ΔΔ* cells. Genes with high RNAPII occupancy are gray shaded, and the corresponding genes are red highlighted. (**e**) Schematic model showing the overlapping and distinct roles of Fun30[Fft3] and FACT in modulating the nucleosome barrier during RNAPII elongation. Fun30[Fft3] travels with elongating RNAPII, probably by associating with RNAPII or transcription elongation factors. During RNAPII elongation, Fun30[Fft3] acts to reduce this barrier by facilitating nucleosome disassembly. FACT cooperates with Fun30[Fft3] to reduce the nucleosome barrier by inducing H2A-H2B dimer disassembly or 'nucleosome breathing' (which occurs without H2A-H2B dimer disassembly). After RNAPII passes, FACT restores the nucleosomes by recycling the original histones. Set2-mediated H3K36me[3] and RNAPII facilitate the recruitment or function of Clr6 complex II, which deacetylates histones behind the elongating RNAPII to restore the stable chromatin state.

*fft3Δ* cells (Fig. 4e). To determine causality, we reduced the nucleosome barrier in *fft3Δ* cells by lowering the levels of histone H3 and H4 and observed a global recovery of RNAPII occupancy at transcribing regions. This demonstrated that increased nucleosome barrier caused by defective nucleosome disassembly at transcribing regions impaired occupancy of RNAPII at transcribing regions in *fft3Δ* cells. (Figs 5 and 6). In our analysis, the main feature that distinguishes Fun30$^{Fft3}$ from other chromatin remodellers is that it is preferentially and transcription-dependently localized to transcribing regions. The Ino80 and RSC complexes, which were also found to affect RNAPII occupancy, were preferentially localized to promoters rather than transcribing regions and are thus likely to regulate RNAPII occupancy mainly through their actions at promoters. However, since the Ino80 and RSC complexes showed some transcription-dependent localization at transcribing regions, we should not completely exclude the possibility that they may modulate the nucleosome barrier at transcribing regions.

A role for Fun30 in transcription was also reported in budding yeast[42]; however, the effect of *fun30Δ* on RNAPII transcription in budding yeast was determined to be marginal. Given the extensive redundancy in the abilities of chromatin remodellers to regulate the nucleosome barrier and transcription initiation at *PHO5* promoter of budding yeast[43], it is possible that the marginal role of budding yeast Fun30 in transcription is due to functional redundancy between Fun30 and other chromatin remodellers. Indeed, recent studies in budding yeast showed that the RSC complex, which promotes transcription through nucleosome-bound templates *in vitro*[25], localizes to coding regions and regulates histone occupancy[26,27]. This suggests that the RSC complex may have a role for modulating the nucleosome barrier at transcribing regions. In this regard, it would be intriguing to investigate whether budding yeast Fun30 plays a redundant role with the RSC complex in regulating RNAPII transcription at coding regions. In humans, the Fun30 homologue, SMARCAD1, was found to be localized to the replicating heterochromatin during late S phase and to have a role in maintaining silent chromatin after DNA replication; however, outside S phase, SMARCAD1 was shown to be widely distributed to the nucleus but not restricted to heterochromatin[44]. Interestingly, human SMARCAD1 associates with the subunits of human FACT complex (SUPT16H and SSRP1) and other chromatin remodelling factors, suggesting a heterochromatin-independent role for SMARCAD1 in chromatin regulation[44]. Furthermore, a recent study demonstrated that *Drosophila* SMARCAD1 facilitates RNAPII transcription both *in vitro* and *in vivo*[45]. These observations, together with our finding that Fun30$^{Fft3}$ cooperates with FACT complex to promote RNAPII transcription through chromatin, suggest that Fun30 may have a conserved role in promoting nucleosome disassembly during RNAPII elongation in higher eukaryotes.

## Methods

**Standard protocols for handling fission yeast cells.** Fission yeast cells were grown until mid-log phase in suitable media following standard protocol[46]. YES was used as rich medium. Fission yeast cells were incubated at 30 °C unless indicated otherwise. To generate the deletion strains and TAP/5xFLAG-tagged strains, we used a PCR-based locus-targeting approach or genetic crossing followed by random spore analysis. Detailed information on the utilized strains is presented in Supplementary Table 2.

**ChIP-seq analysis.** ChIP cells were fixed with 1% formaldehyde for 15 min (for cells grown at 30 °C), 10 min (for cells grown at 37 °C) or 40 min (for cells grown at 22 °C), and then collected. Since high-throughput sequencing requires a sufficient quantity of immunoprecipitated (IPed) DNA to yield an optimal signal, we cultured many more cells for the ChIP-seq experiments than for the conventional ChIP experiments ($4.8 \times 10^8$ cells for H3, RNAPII-CTDun and

RNAPII-CTD$_{S2P}$ ChIP and $1.6 \times 10^9$ cells for FLAG ChIP were used). The amounts of lysis buffers, antibodies, and agarose beads were increased accordingly. Equal amounts of wild-type cells lacking expression of FLAG-tagged proteins were used as tag-free controls. Cell extracts were prepared in lysis buffer (50 mM HEPES-KOH, pH 7.5, 140 mM NaCl, 1 mM EDTA, 1% Triton X-100, 0.1% sodium deoxycholate) using the standard bead-beating method. After IP, the agarose bead-bound IP DNA was washed and eluted on an IP column (SigmaPrep spin column). For FLAG ChIP, we used a modified lysis buffer (50 mM Tris-HCl, pH 7.5, 150 mM NaCl, 1 mM EDTA, 1% Triton X-100) and performed washes using TBS (20 mM Tris-HCl, pH 7.5, 150 mM NaCl) and TBS with 0.05% Tween-20 to prevent the M2 antibody from being denatured by sodium deoxycholate. After de-crosslinking, the DNA samples were purified using a Qiagen PCR purification kit. ChIP-seq libraries were constructed from 10 ng of input DNA or 1–10 ng of IP DNA, using a commercial ChIP-seq kit (NEXTflex ChIP-Seq kit, Bioo Scientific) according to the manufacturer's protocol. For multiplexed libraries, 50-nt single-end reads were sequenced on an Illumina Hiseq 2500. After high-throughput sequencing, the output reads were processed by Trim Galore-driven adapter removal and quality trimming, and aligned to the *S. pombe* ASM294 genome assembly using the Novoalign tool. If a read was mapped to multiple locations, a single location was randomly selected. The reproducibility between biological duplicates was confirmed, and the aligned duplicates were merged using the bamCorrelate tool of deepTools[47]. Each merged read was compared with results from input, wild-type (for H3, RNAPII-CTDun and RNAPII-CTD$_{S2P}$ ChIP) or mock (for FLAG ChIP) using bamCompare, and average metagene analyses were performed using the computeMatrix and plotProfile tools of deepTools. The normalized data were visualized and extracted using the integrative genome viewer (IGV). Further bioinformatic analyses of the ChIP-seq data were performed using the R software package. Spike-in controls were not included in our ChIP-seq samples. Thus, it should be noted that the relative ChIP-seq signals in our analysis may not be sensitive to a global change.

**Co-immunoprecipitation of Fft3-5xFLAG and RNAPII.** Fft3-5xFLAG was purified as previously described with minor modifications[46]. Briefly, a mild lysis buffer (50 mM HEPES, pH 7.6, 75 mM KCl, 1 mM MgCl2, 1 mM EGTA, 0.1% Triton-X100) was used for both lysis and washes. The purified Fft3-5xFLAG, Fft3-5xFLAG, RNAPII-CTD$_{S2P}$ and RNAPII-CTD$_{un}$ proteins were detected by Western blotting. Uncropped versions of the western blots are shown in Supplementary Fig. 7.

**TAP purification.** Fun30$^{Fft3}$ was purified as previously described, with minor modifications[48]. The following buffers were used: Workman buffer containing 40 mM HEPES, pH 7.5, 350 mM NaCl, 10% glycerol, 0.1% Tween-20, 10 mM β-mercapto EtOH, 1 mM PMSF, 1 mM benzamidine, 1 mM pepstatin A, and 1 mM leupeptin; TEV cleavage buffer containing 10 mM Tris-Cl, pH 8.0, 50 mM NaCl, 0.5 mM EDTA, 5 mM MgCl2, 1 mM CaCl2, 0.1% NP-40, 1 mM DTT, 1 mM PMSF, and 1 mM pepstatin A; calmodulin binding buffer containing 10 mM Tris-Cl, pH 8.0, 150 mM NaCl, 1 mM MgAc, 1 mM imidazole, 2 mM CaCl2, 10% glycerol, 0.1% NP-40, 10 mM β-mercapto EtOH, 1 mM PMSF, and 1 mM pepstatin A; and calmodulin elution buffer containing 10 mM Tris-Cl, pH 8.0, 150 mM NaCl, 3 mM EGTA, 1 mM MgAc, 1 mM imidazole, 10% glycerol, 0.05% Tween-20, 10 mM β-mercapto EtOH, and a protease inhibitor cocktail (11873580001, Roche). TAP purification was performed for both TAP-tagged and non-tagged Fun30$^{Fft3}$, and confirmed by silver staining.

**Mathematical model.** To test if transcription-coupled function of Fun30$^{Fft3}$ in nucleosome disassembly at transcribing regions can explain the observed histone H3 occupancy changes at transcribing regions of *fft3Δ* cells, we utilized a formula, $D = L + R$, described in Fig. 4c. By $D$ we denote the steady state level of nucleosomes that are disassembled by RNAPII transcription at transcribing regions relative to total number of nucleosomes at transcribing regions, which is equivalent to the sum of $L$, the steady state level of nucleosomes that are disassembled and not immediately reassembled during RNAPII transcription at transcribing regions relative to total number of nucleosomes at transcribing regions, and $R$, the steady state level of nucleosomes that are transiently disassembled but immediately reassembled during transcription at transcribing regions relative to total number of nucleosomes at transcribing regions. We further defined the parameters $D$, $L$ and $R$ in wild-type, *fft3Δ* and *spt16-18* cells as $D_{wt}$, $D_{fft3Δ}$ and $D_{spt16-18}$ for nucleosome disassembly, $L_{wt}$, $L_{fft3Δ}$ and $L_{spt16-18}$ for nucleosome loss, and $R_{wt}$, $R_{fft3Δ}$ and $R_{spt16-18}$ for nucleosome reassembly, respectively. We formulated a model based on a couple of additional assumptions.

1.$D_{fft3Δ}$ and $D_{spt16-18}$ are proportional to $D_{wt}$.
2.$L_{fft3Δ}$ is proportional to $L_{wt}$.

Assumption 2 led to a formula,

$$L_{fft3\Delta} = x \cdot L_{wt} \ (x : \text{a constant between 0 and 1}; \ 0 \leq x < 1).$$

The levels of histone H3 occupancy changes at transcribing regions in *fft3Δ* cells (equivalent to the levels of nucleosome loss at transcribing regions induced

by Fun30$^{Fft3}$) become,

$$\Delta H3_{fft3\Delta} = \Delta L_{fft3\Delta} = L_{wt} - L_{fft3\Delta}$$
$$= (1-x) \cdot L_{wt} \ (x : \text{a constant between 0 and 1}; \ 0 \leq x < 1).$$

The constant $x$ cannot be 1 as the observed values of $\Delta H3_{fft3\Delta}$ are non-zero. We postulated $R_{spt16-18}$ approximately zero because *spt16-18* cells lack an essential function to reassemble nucleosomes during RNAPII transcription. Given this,

$$D_{spt16-18} = L_{spt16-18} + R_{spt16-18} = L_{spt16-18}.$$

Then the levels of histone H3 occupancy changes at transcribing regions in *spt16-18* cells relative to wild-type cells become,

$$\Delta H3_{spt16-18} = \Delta L_{spt16-18} = L_{wt} - L_{spt16-18} = L_{wt} - D_{spt16-18}.$$

This led to the formula,

$$D_{spt16-18} = L_{wt} - \Delta H3_{spt16-18} = \Delta H3_{fft3\Delta}/(1-x) - \Delta H3_{spt16-18} \ (0 \leq x < 1).$$

The value $1-x$ indicates the decrease in the levels of nucleosome disassembly in *fft3Δ* cells relative to wild-type cells; thus, we defined this as the relative contribution of Fun30$^{Fft3}$ in RNAPII-mediated nucleosome disassembly at transcribing regions. Rewriting the formula according to this definition becomes,

$$D_{spt16-18} = \Delta H3_{fft3\Delta}/x' - \Delta H3_{spt16-18} \ (x', 0 < x' \leq 1),$$

where $x'$ is the relative contribution of Fun30$^{Fft3}$ in RNAPII-mediated nucleosome disassembly at transcribing regions.

Since $D_{spt16-18}$ should be coupled to transcription, the most likely value of $x'$ could be determined by identifying $x'$ ($0 < x' \leq 1$) which results in the maximum correlation between $D_{spt16-18}$ and RNAPII occupancies (Supplementary Fig. 7).

Based on this plot, we could estimate the most likely values of $x'$ to be 84% (r = 0.71) and 54% (r = 0.67) for RNAPII-CTD$_{S2P}$ and RNAPII-CTD$_{un}$, respectively. We could estimate $D_{spt16-18}$ accordingly. By Assumption 1, $D_{wt}$, $D_{fft3\Delta}$ and $D_{spt16-18}$ are proportional to each other. Thus, the levels of Fun30$^{Fft3}$-mediated nucleosome disassembly at transcribing regions ($x' \cdot D_{wt}$) is proportional to $D_{spt16-18}$, which leads to,

$$x' \cdot D_{wt} = k \cdot D_{spt16-18} \ (k, \text{a constant bigger than zero}, k > 0).$$

This allowed us to perform scatterplot analyses to find correlations between the levels of Fun30$^{Fft3}$-mediated nucleosome disassembly at transcribing regions ($x' \cdot D_{wt}$) estimated on an arbitrary scale and RNAPII occupancies as well as RNAPII occupancy changes in *fft3Δ* cells (Fig. 4e).

**Antibodies.** Antibodies against α-RNA polymerase II CTD repeat YSPTSPS without phosphorylation (8WG16; 920102, BioLegend; previously MMS-126R, Covance; 0.1 µg µl$^{-1}$), the α-RNA polymerase II CTD repeat YSPTSPS with phosphorylation at Ser2 (phospho S2; ab5095, Abcam; 1 µg µl$^{-1}$), α-FLAG M2 (F1804, Sigma; 1 µg µl$^{-1}$), and α-beta actin (sc-47778, Santa Cruz; 0.2 µg µl$^{-1}$) were used for ChIP-seq, co-immunoprecipitation and western blot. In addition, the α-H3 antibody (rabbit polyclonal; serum; 1 µg µl$^{-1}$), which was produced in-house against recombinant yeast histone H3[49], were also used for ChIP-seq and western blot.

**Data availability.** The raw and processed sequencing data from this publication have been submitted to the NCBI Gene Expression Omnibus (GEO; http://www.ncbi.nlm.nih.gov/geo/) under accession number GSE83600. Mass spectrometry data can be accessed from the Stowers Original Data Repository at http://www.stowers.org/research/publications/libpb-1085. The authors declare that all the data supporting the findings of this study are available within the article and its Supplementary Information files and from the corresponding author upon reasonable request.

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

## Acknowledgements

We thank Robin Allshire for providing strains and reagents. This work was supported by grants from the Stem Cell Research Program (2012M 3A9B 4027953) and Mid-career Researcher Program (2016R1A2B2006354) through the National Research Foundation of Korea (NRF), and the KAIST Future Systems Healthcare Project funded by the Ministry of Science, ICT, and Future Planning. J.M.G., L.F., M.P.W. and J.L.W. are supported by the Stowers Institute and NIH Grant R35GM118068 to J.L.W.

## Author contributions

J.L., E.S.C. and D.L. conceived and designed the project; J.L. performed most of the experiments and data analyses with inputs from E.S.C. and D.L.; H.D.S. helped purification of Fft3 complex; K.K. contributed to bioinformatic analysis; J.G., L.F., M.W., and J.L.W. performed MudPIT analysis of Fft3 complex; J.L., E.S.C., J.C., J.L.W. and D.L. wrote and modified the manuscript.

## Additional information

**Competing financial interests:** The authors declare no competing financial interests.

