## [Peer Review File · Nature Communications]

Reviewers' comments:

Reviewer #1 (Remarks to the Author):

This is a thorough and high quality analysis of the contribution of the Fun30 chromatin remodeler to nucleosome disassembly within ORFs, and consequently polymerase II occupancy and transcription. The study very nicely takes advantage of the power of combining next generation sequencing analysis of factor occupancy, pol II occupancy, histone occupancy and transcription in a variety of mutants. They use this quite cleverly to arrive at a mechanism and to perform genetics, using next gen sequencing assays as the phenotype. Basically, they find a new function for the FUN30 chromatin remodeler in chromatin disassembly within gene bodies, functioning alongside the histone chaperone FACT, which in turn enables efficient pol II occupancy within ORFs and gene expression. Before this work we did not know which chromatin remodeler helps with histone exchange within ORFs during transcription, so the impact is significant. Experiments are well controlled and accurately interpreted. My suggestions are only minor.

Fig. 1A and 1C: comparisons in the text are made between the results with FACT and FUN30 mutants, however the data for FACT is shown in neither of these panels. Please include spt16 data in Fig. 1A and Spt16 localization data in Fig. 1C.

Figure 4A. Pol II occupancy in the fun30 mutant needs to be included in the screenshot, so that we can see that pol II occupancy specifically goes down as a consequence of failure of Fun30 to remove histones from ORFs like Hsp90.

In Figure 4A, how can H3 occupancy in wild type (green) be negative? same for RNAPol II and Fft3 and Spt16 occupancy. This shows that they have some serious need for normalization controls in their data. Similarly how can histone exchange be negative? Change can be negative, but relative levels of a factor has to be 0 or above. These data need to be corrected.

The description of Fig 4g in the text contradicts what it is actually showing. This is described as Genes at which FACT did not efficiently maintain nucleosomes. however the data is labelled "genes with no decreased H3 in spt16". surely, if they were genes that FACT did not efficiently maintain nucleosomes, they would be "genes with decreased H3 in spt16". This would also make more sense because then the blue in Fig. 4F and 4G would be the same thing, not the opposite of each other.

Reviewer #2 (Remarks to the Author):

In this manuscript, the authors study the Fun30 function in *S. pombe*. Previous work has suggested that Fun30 is a nucleosome remodeler. The experiments presented in this manuscript address the role of Fun30 in controlling the level of RNA polymerase II and histone H3, as well as the relationship between Fun30 and FACT. From their analyses, the authors conclude that Fun30 help to overcome nucleosomal barriers during transcription elongation. There are some interesting results in this paper, but the density of both the writing and the data make the novel results less apparent than they should be. The writing needs some editing to make it more concise and clear, and to shape the story in a way that will make it more accessible to readers. There are also other issues to address, listed below.

Comments are below, generally presented as they arose during the reading of the manuscript.

1. A lot of the conclusions in this work rest on the evaluation of correlation coefficients. In some

cases, it seems that the authors are making a strong conclusion based on r values that may not merit it – the most striking example of this is in Fig. 4b. Compare Fig. 4b ($r=0.41$) and Supp. Fig. 4a ($r=0.25$). The first is discussed as meaningful and the second as not meaningful. Also in many cases, we are asked to evaluate the r values for comparisons between chip-seq of different proteins when we haven't been given the r values for the replicate experiments. We need those values as well. A clearer discussion of what the authors consider to be meaningful r values, as well as more information about duplicates should be provided throughout the manuscript.

2. line 39 – Why cite these references for a statement about epigenetic regulation?

3. lines 42-44 – This statement should be referenced.

4. Introduction – in many cases, the authors have cited specific cases from the primary literature as examples of general statements. For example, on lines 49-50 they have cited two specific examples where the nucleosome remodeling complex RSC plays a role in initiation. While citing primary studies is usually good, in this case it can be misleading, as it neglects a lot of literature on these topics. I suggest for such general statements, that a relevant review should be included. For this particular case, the review by Clapier and Cairns, already cited elsewhere in this manuscript, would be appropriate. One other case concerns the introduction of FACT. While the papers cited are all important, others are not, such as a recent key paper concerning FACT mechanism that has not been cited (Kemble et al., Mol. Cell, 2015).

5. line 106 – In addition to RSC, Swi/Snf has also been shown to play a role in elongation (Schwabish MA, Struhl K., Mol Cell Biol. 2007 Oct;27(20):6987-95). This important paper should be cited.

6. line 140 – The reference cited for CTD phosphorylation (ref. 51) was published back-to-back with another paper (Suh et al.) which should be cited. The two papers were not in agreement on all points, so this should be taken into consideration when discussing published information about CTD phosphorylation.

7. lines 150-151 – The authors chose to use an *fft2 fft3* double mutant based on a single phenotype – cold sensitivity at 25°C. Given the possibility that *Fft1* might make a contribution to function that would not be detectable by checking just one phenotype, I would have strongly recommended that they use the triple mutant so that their study would assess conditions without any Fun30 activity.

8. line 157 – What is meant by high occupancy of RNAPII? How many genes were in that group? More information should be provided.

9. line 448 – The section on antibodies doesn't make clear which one was used for the unphosphorylated IP – was it 8WG16? Please state explicitly.

10. Do any of the mutants studied in Fig. 1b affect RNAPII levels, either unP or S2P? This should be determined to help interpret the ChIP-seq results.

11. line 174 – In contrast to the reference cited about the localization of RSC in *S. cerevisiae*, another paper, (Spain et al. Mol. Cell 2014, 56:653-66) found that RSC was localized across ORFs. This reference is actually cited to make a different point elsewhere in this manuscript.

12. line 177 – Are the differences in the level of *Fft2* and *Fft3* chip explained by the cellular levels of *Fft2* and *Fft3*?

13. Figure 1d – Given that the RNAPII chip-seq signal is most strongly affected in highly transcribed genes, it would be of interest to measure the r values between the different factors

and RNAPII for just that set of genes.

14. Fig 1d: The authors conclude that FACT and Fun30 may function similarly to promote RNAPII occupancy. Is there also a correlation between RNAPII change in occupancy in *spt16* and *fft3* mutants?

15. line 487 – the word “high” is missing

16. line 194 – To test the similarity more directly, the authors should provide the *r* value for the effects on RNAPII.

17. In the experiments where there were temperature shifts, how was the 2 hour time point chosen? Were cell growth and viability measured? This is particularly important for *spt16* mutants, as one way that they were initially identified was as cell cycle mutants (*spt16* is also known as *cdc68*).

18. line 217 – The results are consistent with FACT being required for *fun30+* expression. Has that been tested?

19. lines 242-3 – This sentence should be revised. Looking for a correlation does not determine causation.

20. lines 243-4 – This sentence should be revised. Currently it states that Fft3 chip-seq results were examined in a *fun3-fft3* mutant.

21. line 234: H3K36me3 should not be described as the footprint of elongating RNAPII

22. reference 53 is incomplete

23. line 310 – What happens to expression of the other *fun30* genes in the histone reduction mutant? If they are overexpressed, that could account for the observed phenotypes.

24. Fig. 4 and page 12 – There is a lot of information presented in terms of correlation coefficients and ECDF plots. The authors should provide some specific numbers: for example, in *fft3* mutants, how many genes have increased H3 levels and how many genes do not, and also in the case of comparing histone exchange rate to histone levels in the *fft3* mutant. With that correlation coefficient, and based on the scatterplot, there are many genes that have the opposite effect. This should be discussed.

25. Fig. 4b - How does H3 occupancy change correlate with histone exchange rate at genes with high Fun30 binding?

26. Fig. 4d - The correlation between H3 occupancy change and RNAPII occupancy change does not show that Fun30 is causally related to the regulation of RNAPII occupancy (Line 261).

27. Fig. 5b - The authors could discuss why they see an increase in histone H3 occupancy at gene bodies in the H3/H4 Δ mutant.

28. A brief introduction of what the function of Pst1 and Cph1 is in the Clr6 complex would be helpful. Are they the catalytic subunits?

29. lines 376-378 - A reference should be cited.

30. Several recent sets of studies have provided evidence that spike-in controls are important when measuring the genome-wide effects of potentially global factors. For example, see Chen et

al. (MCB 36, 662) and Loven et al. (Cell 151, 476). As those weren't included in these experiments, the authors should consider mentioning that as a caveat.

Reviewer #3 (Remarks to the Author):

This study by Lee et al, addresses the role of Fun30 in regulating cotranscriptional histone occupancy and promoting transcription in fission yeast. They show that cells lacking the two of the three Fun30 paralogs, Ff2 and Fft3, elicit reduced RNA polymerase II (Pol II) occupancy in coding regions. Consistent with this, they also report these two subunits localize to coding sequences of strongly transcribed genes. Thus, their data clearly implicate Fun30 in promoting transcription during Pol II elongation. Their study further shows that Fun30 might collaborate with the histone chaperone FACT to modulate histone occupancy and to promote transcription. The role of Fun30 in promoting cotranscriptional histone disassembly is further supported by their data showing that reducing histone occupancy in the cells bypasses the requirement of Fun30, to a certain extent. Overall, this is a very well done study and the findings reported here will be of interest to transcription and chromatin field in general.

One of the major concerns is that their claim of Fun30 collaborating with FACT is not well supported by the data presented in this study. They state that both Fun30 and FACT display good correlation with Pol II occupancy, but this will be true if they analyze the correlation with other elongation factors such as Spt6 or the PAF complex. Thus, this only suggests that Fun30 occupancy is correlated with transcription, and not particularly with FACT. Although, their data showing that significant number of genes are impaired for transcription in both *spt16-1* and *fft3Δ* strains lend support to their idea, they could further bolster it by showing that it is not true for other elongation factors.

Other comments:

Figure 1.

1. Authors should provide the number of genes included in the metagene analysis (Fig. 1a and 1c).
2. Although, authors show enrichment of Fun30 in coding sequences correlates with transcription, have the authors performed cluster analysis to examine if there are a considerable number of genes those display high Pol II occupancy but lacks Fun30 enrichment in ORFs?
3. Did the authors find that both Fft2 and Fft3 are recruited to same gene-sets or do they occupy different sets of genes?
4. The authors show changes in Pol II occupancy in *fft3Δ* strains. Did they measure this also at 37 degree, as they did for the FACT mutants? This is particularly important considering recent genome-wide studies showing global changes in transcription when cells are exposed to heat-shock. Thus, it is possible that some of the differences observed by the authors may be attributed to the heat-shock treatment.

Figure 2.

In this figure, the authors investigated Pol II occupancy in the *fft3Δ* and *spt16-1* single and *fft3Δ/spt16-1* double mutant to evaluate possible cooperation between remodelers and histone chaperone. Although, growth assay performed at the permissive temperature showed a synthetic growth phenotype, the effect of the double mutant on Pol II occupancy was very similar to that of the *spt16-1* single mutant. Thus, it appears that FACT may play a dominant role in regulating transcription, and the impact of lacking Fun30 is masked by the severe phenotype displayed by the FACT mutant. Their analysis identified a significant number of genes displaying Pol II occupancy defect $< -0.5 \log_2$ both in *fft3Δ* and FACT mutant. However, their analysis also suggests that there are genes, which elicit greater reduction on deleting Fft3 than impairing Spt16. Have the authors analyzed whether the genes showing reduced Pol II occupancy in *fft3Δ* are indeed Fft3 targets. In other words, what is the level of enrichment of Fft3 at these genes in WT cells?

Figure 4.

1. In this figure, the authors have compared the changes in H3 occupancy in the Fun30 mutant with the histone exchange over ORFs and found these to be positively correlated. They further suggest that the reduction in Pol II occupancy in fun30 mutant is caused by defective nucleosome occupancy. This is primarily based on analysis carried out on genes showing higher histone exchange. It is difficult to assess the degree of changes in H3 occupancy or Pol II occupancy that occurs in mutant. The authors should explicitly mention how they choose these genes, and how much defect in H3 occupancy was observed at these genes in the ftt3 Δ mutant.

2. They should also specify whether the genes undergoing histone exchange were among those which showed Fun30 recruitment.

3. (Page 13, line 266) The authors mention that "the reduced RNAPII occupancy defect... among those that undergo rapid histone turnover". And later (page 14, lines 287-291) they suggest that Fun30-triggered a nucleosome loss may have been underrepresented at many genes due to efficient reassembly by the FACT complex. The latter statement implies that there is active histone disassembly and reassembly, and thus what authors may be looking at in their histone exchange data is the integration of newly synthesized histones rather than the ones which are removed during transcription. This will also fit with previous observations that highest loss of histones is generally seen at strongly transcribed ORFs.

Figure 5.

Suppression of the cold-sensitive phenotype by deletion of two of the three H3/H4 copies strongly supports authors claim that Fun30 is involved in nucleosome disassembly. However, it is puzzling to see that slightly higher histone occupancy is observed in coding regions of genes included in Fig 5b. One would expect the ratio (mutant/WT) to be lower, considering that both histone levels and histone occupancies are reduced in H3/H4 $\Delta\Delta$ strain (suppl. Fig 5). Could authors comment on this observation? Do the authors see significant changes in histone occupancy in H3/H4 $\Delta\Delta$ vs WT strain?

Point-by-point response to reviewer's comments

Reviewer #1 (Remarks to the Author):

This is a thorough and high quality analysis of the contribution of the Fun30 chromatin remodeler to nucleosome disassembly within ORFs, and consequently polymerase II occupancy and transcription. The study very nicely takes advantage of the power of combining next generation sequencing analysis of factor occupancy, pol II occupancy, histone occupancy and transcription in a variety of mutants. They use this quite cleverly to arrive at a mechanism and to perform genetics, using next gen sequencing assays as the phenotype. Basically, they find a new function for the FUN30 chromatin remodeler in chromatin disassembly within gene bodies, functioning alongside the histone chaperone FACT, which in turn enables efficient pol II occupancy within ORFs and gene expression. Before this work we did not know which chromatin remodeler helps with histone exchange within ORFs during transcription, so the impact is significant. Experiments are well controlled and accurately interpreted. My suggestions are only minor.

*Specific comment: Fig. 1A and 1C: comparisons in the text are made between the results with FACT and FUN30 mutants, however the data for FACT is shown in neither of these panels. Please include *spt16* data in Fig. 1A and *Spt16* localization data in Fig. 1C.*

Our Response: RNAPII occupancy changes of *spt16-18* cells appear overly dramatic compared with those of chromatin remodeler mutants and thus make comparisons between the chromatin remodeler

mutants less feasible when included in **Fig. 1a** (see **fig. a**). The same is true for ChIP-seq signals of Spt16-GFP at ORFs which are too high to be compared with those of chromatin remodelers in the same metagene profile (see **fig. b**). For this reason, we decided to keep the original **Fig. 1a** and **Fig. 1c** which separated Spt16 data from those of chromatin remodelers.

***Specific comment:** Figure 4A. Pol II occupancy in the *fun30* mutant needs to be included in the screenshot, so that we can see that pol II occupancy specifically goes down as a consequence of failure of *Fun30* to remove histones from ORFs like *Hsp90*.*

Our Response: We also deeply thought about Figure 4 and decided that we would revise figure 4 clearly to address reviewers' concerns. In the revised manuscript, we now introduced a mathematical model to estimate the levels of nucleosome disassembly by $\text{Fun30}^{\text{Fft3}}$ at ORFs based on the available experimental data (see **Fig. 4c** and **Mathematical model** in **Methods** in the revised manuscript). We performed scatterplot analyses according to the estimation and found that the levels of nucleosome disassembly by $\text{Fun30}^{\text{Fft3}}$ at ORFs correlate with RNAPII occupancy changes in *fft3Δ* cells as well as RNAPII occupancies among the whole protein-coding genes (see **Fig. 4d,e** in the revised manuscript). Thus, selection of a specific subset of genes was no longer required to show the correlation between the function of $\text{Fun30}^{\text{Fft3}}$ in nucleosome disassembly at ORFs and the function of $\text{Fun30}^{\text{Fft3}}$ in regulation of RNAPII occupancy. We therefore removed some of the figures related to selection of the specific subset of genes (**Fig. 4a,c,d,e,f,g** in the original manuscript) and replaced them with new figures (**Fig. 4c,d,e** in the revised manuscript). We accordingly modified the main text, which significantly improved the conclusion.

***Specific comment:** In Figure 4A, how can H3 occupancy in wild type (green) be negative? same for RNAPol II and *Fft3* and *Spt16* occupancy. This shows that they have some serious need for normalization controls in their data. Similarly how can histone exchange be negative? Change can be negative, but relative levels of a factor has to be 0 or above. These data need to be corrected.*

Our Response: We obtained ChIP-seq enrichment by calculating $\log_2(\text{IP} / \text{input})$. To compensate for differences in mapping efficiency and sequencing depth among the ChIP-seq samples, the conventional ChIP-seq analysis normalizes readcounts of IP or input sample at a genomic locus (calculated per 10 bp bin in our analysis) relative to total readcounts of each sample. Thus, the normalized signals of IP at a given locus can be smaller than those of input (i.e. $\text{IP} < \text{input}$), and in this case $\log_2(\text{IP} / \text{input})$ becomes negative.

Specific comment: The description of Fig 4g in the text contradicts what it is actually showing. This is described as genes at which FACT did not efficiently maintain nucleosomes. However the data is labelled "genes with no decreased H3 in spt16". surely, if they were genes that FACT did not efficiently maintain nucleosomes, they would be "genes with decreased H3 in spt16". This would also make more sense because then the blue in Fig. 4F and 4G would be the same thing, not the opposite of each other.

Our Response: As described earlier (page 3), we improved our analysis on nucleosome disassembly at ORFs by Fun30^{Fit3} using a simple mathematical model. According to our analysis, nucleosome loss at ORFs by Fun30^{Fit3} may not be coupled to transcription even if nucleosome disassembly at ORFs by Fun30^{Fit3} is coupled to transcription because only some of the disassembled nucleosomes are lost and the rest of them are efficiently reassembled by nucleosome reassembly pathway such as FACT. We included methods and results regarding this analysis (**Mathematical model in Methods, Fig. 4c,d,e**) in the revised manuscript which replaced the original **Fig. 4a,c,d,e,f,g**.

Reviewer #2 (Remarks to the Author):

In this manuscript, the authors study the Fun30 function in S. pombe. Previous work has suggested that Fun30 is a nucleosome remodeler. The experiments presented in this manuscript address the role

of Fun30 in controlling the level of RNA polymerase II and histone H3, as well as the relationship between Fun30 and FACT. From their analyses, the authors conclude that Fun30 help to overcome nucleosomal barriers during transcription elongation.

General comment: *There are some interesting results in this paper, but the density of both the writing and the data make the novel results less apparent than they should be. The writing needs some editing to make it more concise and clear, and to shape the story in a way that will make it more accessible to readers. There are also other issues to address, listed below.*

Our Response: According to the reviewer's suggestions, we edited the manuscript in a way to make it more concise and clear. Especially, we removed the first part of **INTRODUCTION** section which describes the general role of nucleosomes in DNA transactions as well as in preventing transcription initiation at promoters as these are not directly related to the main focus of our study.

Comments are below, generally presented as they arose during the reading of the manuscript.

Specific comment: *1. A lot of the conclusions in this work rest on the evaluation of correlation coefficients. In some cases, it seems that the authors are making a strong conclusion based on r values that may not merit it – the most striking example of this is in Fig. 4b. Compare Fig. 4b ($r=0.41$) and Supp. Fig. 4a ($r=0.25$). The first is discussed as meaningful and the second as not meaningful. Also in many cases, we are asked to evaluate the r values for comparisons between chip-seq of different proteins when we haven't been given the r values for the replicate experiments. We need those values as well. A clearer discussion of what the authors consider to be meaningful r values, as well as more information about duplicates should be provided throughout the manuscript.*

Our Response: We interpreted the correlation coefficients greater than 0.4 ($r \geq 0.4$) to indicate a positive correlation according to a widely-used interpretation of the correlation coefficients in statistics. Generally, r value 0.00 ~ 0.19 is interpreted as very weak, 0.20 ~ 0.39 to be weak, 0.40 ~

0.59 to be moderate, 0.60 ~ 0.79 to be strong, and 0.80 ~ 1.00 to be very strong (Evans, James D. Straightforward statistics for the behavioral sciences. *Brooks/Cole*, 1996.). Based on this interpretation, the correlation coefficients less than 0.4 ($r < 0.4$) are weak or very weak, so that we considered them not sufficient to indicate a positive correlation. We explicitly described this in the main text of revised manuscript: “We considered the absolute values of correlation coefficients greater than 0.4 ($r \geq 0.4$) as a meaningful correlation”. In addition to this, when we merged ChIP-seq data of biological duplicates we validated that they are sufficiently correlative (the minimum r between duplicates: 0.92). Below, we added a table showing the correlation coefficients determined between ChIP-seq data of biological duplicates.

r value of biological duplicates
bam files of biological duplicates were compared by bamCorrelate
r value was calculated by pearson correlation method of plotCorrelation
n.a means there is no file for that category
One with higher enrichment was selected for H3K36me3, Spt6_HA and Set2_myc of GSE49574

screening	RNAPII_CTDS2P	RNAPII_CTDun	H3	Factor binding	input
Wild type_30C	0.99	0.97	n.a	n.a	1
fft1D_30C	0.99	0.99	n.a	n.a	n.a
fft2D_30C	1	1	n.a	n.a	n.a
fft3D_30C	0.99	0.99	n.a	n.a	n.a
fft2Dfft3D_30C	0.99	0.99	n.a	n.a	n.a
hrp1D_30C	0.99	0.99	n.a	n.a	n.a
hrp3D_30C	0.99	0.99	n.a	n.a	n.a
ies6D_30C	0.99	0.99	n.a	n.a	n.a
snf22D_30C	0.99	0.99	n.a	n.a	n.a
swr1D_30C	0.99	1	n.a	n.a	n.a
wild type_37C	1	0.98	0.92	n.a	n.a
spt16-18_37C	0.96	0.99	0.96	n.a	n.a
snf21-36_37C	0.99	0.99	n.a	n.a	n.a
M2_mock	n.a	n.a	n.a	1	n.a
fft2_FLAG	n.a	n.a	n.a	0.99	n.a
fft3_FLAG	n.a	n.a	n.a	1	n.a
ino80_FLAG	n.a	n.a	n.a	1	n.a
snf21_FLAG	n.a	n.a	n.a	1	n.a
Spt16_GFP	n.a	n.a	n.a	0.99	n.a

Fft3Dxspt16-1	RNAPII_CTDS2P	RNAPII_CTDun	H3	input
wild type_37C	0.99	0.99	0.98	1
fft3D_37C	0.99	0.96	0.98	n.a
spt16-1_37C	0.99	0.99	n.a	n.a
fft3D spt16-1_37C	0.98	0.96	n.a	n.a

fft3D x H3H4DD	RNAPII_CTDS2P	RNAPII_CTDun	H3	input
wild type_22C	1	0.97	0.98	1
fft3D_22C	0.99	0.98	0.99	n.a
H3H4DD_22C	1	1	0.98	n.a
fft3D H3H4DD_22C	1	0.99	0.99	n.a

Specific comment: 2. line 39 – Why cite these references for a statement about epigenetic regulation?

Our Response: Thank you for your comment. However, since this statement was deleted during introduction editing for compacting manuscript, the references were deleted in revised manuscript.

Specific comment: 3. lines 42-44 – This statement should be referenced.

Our Response: Since this statement was also deleted during introduction editing for compacting manuscript, the references were deleted in revised manuscript.

Specific comment: 4. Introduction – in many cases, the authors have cited specific cases from the primary literature as examples of general statements. For example, on lines 49-50 they have cited two specific examples where the nucleosome remodeling complex RSC plays a role in initiation. While citing primary studies is usually good, in this case it can be misleading, as it neglects a lot of literature on these topics. I suggest for such general statements, that a relevant review should be included. For this particular case, the review by Clapier and Cairns, already cited elsewhere in this manuscript, would be appropriate. One other case concerns the introduction of FACT. While the papers cited are all important, others are not, such as a recent key paper concerning FACT mechanism that has not been cited (Kemble et al., Mol. Cell, 2015).

Our Response: Thank you for your comment. Since the introduction to the role of RSC in transcription initiation has been removed, the review is not referenced. But, we modified the reference for FACT mechanism according to the reviewer's suggestion.

Specific comment: 5. line 106 – In addition to RSC, Swi/Snf has also been shown to play a role in elongation (Schwabish MA, Struhl K., Mol Cell Biol. 2007 Oct;27(20):6987-95). This important paper should be cited.

Our Response: We added a new reference according to the reviewer's suggestion.

Specific comment: 6. line 140 – The reference cited for CTD phosphorylation (ref. 51) was published back-to-back with another paper (Suh et al.) which should be cited. The two papers were not in agreement on all points, so this should be taken into consideration when discussing published information about CTD phosphorylation.

Our Response: We added a new reference according to the reviewer's suggestion.

Specific comment: 7. lines 150-151 – The authors chose to use an *fft2 fft3* double mutant based on a single phenotype – cold sensitivity at 25°C. Given the possibility that *Fft1* might make a contribution to function that would not be detectable by checking just one phenotype, I would have strongly recommended that they use the triple mutant so that their study would assess conditions without any *Fun30* activity.

Our Response: To address the issue raised by the reviewer, we performed spotting assays using cells bearing all possible combinations of *fft1Δ*, *fft2Δ* and *fft3Δ* (**Supplementary Fig. 1** in the revised manuscript). In addition, we grew cells under various stress conditions such as 6-Azauracil (blocking growth in transcription elongation defective mutant cells), cadmium sulfate (causing oxidative stress) and potassium chloride and sorbitol (causing hyper-osmotic shock) as well as cold temperature. We found that *fft2Δ* did not cause any phenotype on its own but exacerbates growth defects of *fft3Δ* cells when combined. However, *fft1Δ* did not cause any growth defect under normal and stress conditions alone and even when combined with mutations in other *Fun30* paralogs. This suggests that *Fft1* does not perform any noticeable function required for cell survival and transcription or *Fft1* functions in a manner dependent on *Fft2* or *Fft3*. Based on this observation, we concluded that *fft2Δ fft3Δ* cells can represent cells lacking *Fun30* function in fission yeast and discussed this point in the main text.

Specific comment: 8. line 157 – What is meant by high occupancy of RNAPII? How many genes were in that group? More information should be provided.

Our Response: To categorize RNAPII occupancies at ORFs into 3 classes (high, moderate and low), the RNAPII-CTD_{S2P} and RNAPII-CTD_{un} occupancy profiles in wild-type cells grown at 30°C or 37°C were individually subject to k-means clustering (k = 3). We described this in the legend of Fig. 1a and accordingly labelled the number of genes in each class (high, moderate and low) when they appear in the figures (**Fig. 1a,c**, **Fig. 4a** and **Supplementary Fig. 2a,b**).

Specific comment: 9. line 448 – The section on antibodies doesn't make clear which one was used for the unphosphorylated IP – was it 8WG16? Please state explicitly.

Our Response: We are very sorry for confused description about antibody. In the section on antibodies in **Methods** of the revised manuscript, we explicitly stated that we used antibodies against α -RNA polymerase II CTD repeat YSPTSPS without phosphorylation (8WG16; 920102, BioLegend; previously MMS-126R, Covance) and the α -RNA polymerase II CTD repeat YSPTSPS with phosphorylation at Ser2 (phospho S2; ab5095, Abcam).

Specific comment: 10. Do any of the mutants studied in Fig. 1b affect RNAPII levels, either unP or S2P? This should be determined to help interpret the ChIP-seq results.

Our Response: To address the issue raised by the reviewer, we performed western blot analysis of RNAPII. We found that the level of RNAPII-CTD_{un} is not significantly affected in any of the mutants used in **Fig. 1b**, suggesting that chromatin remodelers and FACT may not have affected RNAPII occupancies at ORFs indirectly by altering RNAPII levels (see below). We also found that the level of RNAPII-CTD_{S2P} is not significantly affected in *fft2Δ fft3Δ* cells, suggesting that Fun30 paralogs did not affect RNAPII occupancies by altering RNAPII-CTD phosphorylation. We noticed that mutations in FACT and other chromatin remodelers have some effects on the level of RNAPII-CTD_{S2P},

suggesting that they may somehow affect RNAPII-CTD phosphorylation. However, we wished to focus on the role of Fun30 paralogs, and thus did not discuss this point in the revised manuscript.

Specific comment: 11. line 174 – In contrast to the reference cited about the localization of RSC in *S. cerevisiae*, another paper, (Spain et al. *Mol. Cell* 2014, 56:653-66) found that RSC was localized across ORFs. This reference is actually cited to make a different point elsewhere in this manuscript.

Our Response: We modified the reference according to the reviewer’s suggestion.

Specific comment: 12. line 177 – Are the differences in the level of Fft2 and Fft3 chip explained by the cellular levels of Fft2 and Fft3?

Our Response: The differences in the level of Fft2 and Fft3 association with chromatin can be explained by the differences in their relative cellular levels if Fft2 and Fft3 have the same chromatin binding properties. In this regard, we compared cellular levels of Fft2 and Fft3 as well as the genomic localizations of Fft2 and Fft3. Western blot analysis revealed that Fft3 is ~8 times more abundant than Fft2 (see below), which may indicate that the relatively high abundance of Fft3 is attributable to the higher association of Fft3 with ORFs (see below).

However, we found that Fft2 and Fft3 have some degree of specificities in their genomic localizations. For example, unlike Fft3, which is preferentially localized to ORFs, Fft2 is significantly enriched also at promoters and terminators (**Fig. 1c** and **Supplementary Fig. 2b** in the revised manuscript). In addition, consistent with a recent report (Persson, Jenna, et al. "Regulating retrotransposon activity through the use of alternative transcription start sites." *EMBO reports* (2016): e201541866), we detected a specific enrichment of Fft2 at retrotransposon-flanking long terminal repeat (LTR) elements (**Supplementary Fig. 2c** in the revised manuscript). Thus, the differences in chromatin association of Fft2 and Fft3 may not be explained solely by the differences in their relative cellular levels. We described the differences in genomic localizations of Fft2 and Fft3 in more detail (main text of the revised manuscript and **Supplementary Fig. 2c**).

Specific comment: 13. *Figure 1d – Given that the RNAPII chip-seq signal is most strongly affected in highly transcribed genes, it would be of interest to measure the r values between the different factors and RNAPII for just that set of genes.*

Our Response: The correlations between RNAPII occupancy and the enrichment of chromatin remodelers and FACT at ORFs were indeed improved to some extent when the correlations were tested among the highly transcribed genes (see below). However, even in this subset of genes, enrichments of Fun30 paralogs Fft2 and Fft3 show higher correlations with RNAPII occupancies than enrichments of other chromatin remodelers such as Ino80 and Snf21 do. This indicates that the main conclusion of **Fig. 1d** was not significantly changed even when we select a specific subset of genes.

We decided not to include the analysis of this specific subset because we wished to show the genome-wide functions of Fun30 paralogs.

Specific comment: 14. Fig 1d: The authors conclude that FACT and Fun30 may function similarly to promote RNAPII occupancy. Is there also a correlation between RNAPII change in occupancy in *spt16* and *fft3* mutants?

Our Response: We appreciate the helpful suggestion. To compare the functions of FACT and Fun30 paralogs in the regulation of RNAPII occupancy more directly, we performed ChIP-seq analysis of RNAPII in cells lacking FACT function (*spt16-18*) and cells lacking Fun30 function (*fft2Δ fft3Δ*) grown at the restrictive temperature of *spt16-18* cells (37°C). As shown in the figure below (included as **Supplementary Fig. 2d** in the revised manuscript), we found that RNAPII occupancy changes of *spt16-18* cells positively correlate with those of *fft2Δ fft3Δ* cells. Inspired by this suggestion, we also directly compared the enrichment of FACT (Spt16-GFP) and Fun30 paralogs (Fft2-5xFLAG and Fft3-5xFLAG) at ORFs, which revealed positive correlations between their localizations to ORFs (**Supplementary Fig. 2e** in the revised manuscript). This suggests that that Fun30 paralogs localize to

ORFs and regulate RNAPII occupancy in a manner similar to FACT. We discussed this point in the main text together with new figures (**Supplementary Fig. 2d,e**) of the revised manuscript.

Specific comment: 15. line 487 – the word “high” is missing

Our Response: We added to the word “high” according to the reviewer’s comment.

Specific comment: 16. line 194 – To test the similarity more directly, the authors should provide the r value for the effects on RNAPII.

Our Response: As described in the response to specific comment #14, the r values of correlations between RNAPII occupancy changes of *fft2Δfft3Δ* cells and those of *spt16-18* cells are 0.42 for RNAPII-CTD_{S2P} and 0.50 for RNAPII-CTD_{un}. We included this analysis as **Supplementary Fig. 2d** of the revised manuscript and described about it in the main text.

Specific comment: 17. In the experiments where there were temperature shifts, how was the 2 hour time point chosen? Were cell growth and viability measured? This is particularly important for *spt16* mutants, as one way that they were initially identified was as cell cycle mutants (*spt16* is also known as *cdc68*).

Our Response: We have chosen 2 hour time point on the basis of the observation that *spt16-1* cells and *spt16-18* cells grow normally as wild-type cells up to 2 hours after temperature shift to 37°C (see below; **figure a**). We noticed that *spt16-18* cells show more severe growth retardation than *spt16-1* and the wild-type cells after 2 hours at 37°C. Thus, the secondary effects caused by defective FACT function would accumulate much faster in *spt16-18* cells than in *spt16-1* cells after 2 hours at 37°C. However, we found that the RNAPII occupancy changes in *spt16-18* cells are nearly identical to those of *spt16-1* cells when measured at 2 hours after temperature shift to 37°C (see below; **figure b**), suggesting that the RNAPII occupancy changes in these mutants at this time point were not significantly affected by the secondary effects.

Specific comment: 18. line 217 – The results are consistent with FACT being required for *fun30+* expression. Has that been tested?

Our Response: To solve the reviewer’s concern, we performed Western blot analysis of Fft3-5xFLAG in wild-type cells and *spt16-18* cells (grown at 37°C for 2 hours after temperature shift from 25°C) and found that Fft3 expression is not significantly affected by *spt16-18* (see below). This excludes the possibility that functional overlap between FACT and Fft3 is attributable to an indirect effect of *spt16-18* affecting Fft3 expression.

Specific comment: 19. lines 242-3 – This sentence should be revised. Looking for a correlation does not determine causation.

Our Response: We substantially modified Fig. 4 and rewrote the main text according to new figures (described in our responses to the reviewer 1’s comments). According to the reviewer’s suggestion, in the revised manuscript we removed the incorrect usage of correlations to infer causation.

Specific comment: 20. lines 243-4 – This sentence should be revised. Currently it states that Fft3 chip-seq results were examined in a *fun3-fft3* mutant.

Our Response: During editing this section (**Fig. 4** and **Supplementary Fig. 4**), that sentence is not included in the revised manuscript.

Specific comment: 21. line 234: H3K36me3 should not be described as the footprint of elongating RNAPII

Our Response: We modified the description of H3K36me3 from “the footprint of elongating RNAPII” to “Set2-mediated H3K36me3” according to the reviewer’s suggestion in the revised manuscript.

Specific comment: 22. reference 53 is incomplete

Our Response: We modified the reference according to the reviewer's suggestion.

Specific comment: 23. line 310 – What happens to expression of the other *fun30* genes in the histone reduction mutant? If they are overexpressed, that could account for the observed phenotypes.

Our Response: To address this issue, we compared the levels of Fft2-5xFLAG in wild-type and histone copy-number mutants by Western blotting (see below). In this analysis, we found that Fun30^{Fft2} expression is not significantly affected by deletion of histone genes, suggesting that the suppression of *fft3Δ* phenotypes by *H3/H4ΔΔ* did not result from overexpression of Fun30^{Fft2}.

Specific comment: 24. Fig. 4 and page 12 – There is a lot of information presented in terms of correlation coefficients and ECDF plots. The authors should provide some specific numbers: for example, in *fft3* mutants, how many genes have increased H3 levels and how many genes do not, and also in the case of comparing histone exchange rate to histone levels in the *fft3* mutant. With that correlation coefficient, and based on the scatterplot, there are many genes that have the opposite effect. This should be discussed.

Our Response: As described earlier, we introduced a mathematical model to estimate the levels of nucleosome disassembly by Fun30^{Fit3} at ORFs based on the available experimental data (see **Fig. 4c** and **Mathematical model** in **Methods** in the revised manuscript). We performed scatterplot analyses according to the estimation and found that the levels of nucleosome disassembly by Fun30^{Fit3} at ORFs correlate with RNAPII occupancy changes in *fft3Δ* cells as well as RNAPII occupancies among the whole protein-coding genes (see **Fig. 4d,e** in the revised manuscript). Thus, selection of a specific subset of genes was no longer required to show the correlation between the function of Fun30^{Fit3} in nucleosome disassembly at ORFs and the function of Fun30^{Fit3} in regulation of RNAPII occupancy. We therefore removed some of the figures related to selection of the specific subset of genes (**Fig. 4a,c,d,e,f,g** in the original manuscript) and replaced them with new figures (Fig. 4c,d,e in the revised manuscript). We accordingly modified the main text. With regard to the correlation between the rate of histone exchange and histone H3 occupancy changes in *fft3Δ* cells, the correlation is limited ($r = 0.41$) because the rate of histone exchange is measured by the rate of new histone incorporation, thereby linked to nucleosome loss only in an indirect manner. It is possible that nucleosome loss at ORFs does not always cause incorporation of new histones.

Specific comment: 25. Fig. 4b - How does H3 occupancy change correlate with histone exchange rate at genes with high Fun30 binding?

Our Response: The correlation between histone exchange rate and histone H3 occupancy change in *fft3Δ* cells at genes with high Fun30^{Fit3} occupancy ($r = 0.43$) was stronger than that at genes with low Fun30^{Fit3} occupancy ($r = 0.34$). However, the correlation coefficient determined among genes with high Fun30^{Fit3} enrichment ($r = 0.43$) was not significantly higher than that determined among the whole protein coding genes ($r = 0.41$; **Fig. 4b**). Thus, we decided to maintain the original **Fig. 4b**.

Specific comment: 26. Fig. 4d - The correlation between H3 occupancy change and RNAPII occupancy change does not show that *Fun30* is causally related to the regulation of RNAPII occupancy (Line 261).

Our Response: As described earlier, we removed the incorrect usage of correlations to infer causation in the revised manuscript according to the reviewer's suggestion.

Specific comment: 27. Fig. 5b - The authors could discuss why they see an increase in histone H3 occupancy at gene bodies in the *H3/H4ΔΔ* mutant.

Our Response: Thank you for making us aware of this issue. Owing to the reviewer's comment, we noticed that there was a mistake in the processing of this particular ChIP-seq experiment, more specifically during bigwig file generation using Deeptools. We corrected this and found that histone H3 occupancies were clearly reduced at ORFs in the *H3/H4ΔΔ* mutant. We accordingly modified **Fig. 5b** in the revised manuscript.

Specific comment: 28. A brief introduction of what the function of *Pst1* and *Cph1* is in the *Clr6* complex would be helpful. Are they the catalytic subunits?

Our Response: According to the reviewer's suggestion, we included a brief introduction to Pst1 and Cph1 in the revised manuscript as follows. "Subunits of Clr6 complex II are non-essential except the catalytic subunit Clr6. A subunit of Clr6 complex II, Cph1 is a fission yeast homolog of budding yeast Rco1, a PHD zinc-finger domain containing protein of Rpd3S complex. We found that *cph1Δ* suppressed the growth defects of *fft2Δ fft3Δ* cells and the cold sensitivity of *fft3Δ* cells (**Fig. 5c**). Another subunit Pst2 is a fission yeast homolog of Sin3 proteins which are conserved from budding yeast to mammals."

Specific comment: 29. lines 376-378 - A reference should be cited.

Our Response: We added a new reference regarding this according to the reviewer's suggestion.

Specific comment: 30. Several recent sets of studies have provided evidence that spike-in controls are important when measuring the genome-wide effects of potentially global factors. For example, see Chen et al. (*MCB* 36, 662) and Loven et al. (*Cell* 151, 476). As those weren't included in these experiments, the authors should consider mentioning that as a caveat.

Our Response: According to the reviewer's suggestion, we mentioned about this in the ChIP-seq analysis section of **Methods** as follows. "Spike-in controls were not included in our ChIP-seq samples. Thus, it should be noted that the ChIP-seq signals in our analysis represent a relative rather than an absolute measurement of enrichment."

Reviewer #3 (Remarks to the Author):

This study by Lee et al, addresses the role of Fun30 in regulating co-transcriptional histone occupancy and promoting transcription in fission yeast. They show that cells lacking the two of the three Fun30 paralogs, Fft2 and Fft3, elicit reduced RNA polymerase II (Pol II) occupancy in coding

regions. Consistent with this, they also report these two subunits localize to coding sequences of strongly transcribed genes. Thus, their data clearly implicate Fun30 in promoting transcription during Pol II elongation. Their study further shows that Fun30 might collaborate with the histone chaperone FACT to modulate histone occupancy and to promote transcription. The role of Fun30 in promoting co-transcriptional histone disassembly is further supported by their data showing that reducing histone occupancy in the cells bypasses the requirement of Fun30, to a certain extent. Overall, this is a very well done study and the findings reported here will be of interest to transcription and chromatin field in general.

General comment: *One of the major concerns is that their claim of Fun30 collaborating with FACT is not well supported by the data presented in this study. They state that both Fun30 and FACT display good correlation with Pol II occupancy, but this will be true if they analyze the correlation with other elongation factors such as Spt6 or the PAF complex. Thus, this only suggests that Fun30 occupancy is correlated with transcription, and not particularly with FACT. Although, their data showing that significant number of genes are impaired for transcription in both *spt16-1* and *fft3Δ* strains lend support to their idea, they could further bolster it by showing that it is not true for other elongation factors.*

Our Response: We appreciate the constructive suggestion. To examine the functional relationships between Fun30^{Fit3} and transcription elongation factors further, we tested genetic interactions of *fft3Δ* with mutations in other transcription elongation-related factors such as Spt6 and Paf1 (*spt6-1* and *paf1Δ*) in addition to mutations in FACT (*spt16-1* and *spt16-2*). Spotting assay revealed that *fft3Δ* does not cause a synthetic lethality when combined with either *spt6-1* or *paf1Δ*, suggesting that Fun30^{Fit3} shares an essential function with FACT but not apparently with Spt6 and Paf1 (**Fig. 2a** in the revised manuscript). We included the new spotting assays in **Fig. 2a** and modified the main text accordingly.

Other comments:

Figure 1.

Specific comment: 1. Authors should provide the number of genes included in the metagene analysis (Fig. 1a and 1c).

Our Response: According to the reviewer's suggestion, we labelled the number of genes in **Fig. 1**.

Specific comment: 2. Although, authors show enrichment of Fun30 in coding sequences correlates with transcription, have the authors performed cluster analysis to examine if there are a considerable number of genes those display high Pol II occupancy but lacks Fun30 enrichment in ORFs?

Our Response: To address the reviewer's question, we performed Venn diagram analysis of the overlap between ORFs with high RNAPII occupancy and those with high or low enrichment of Fun30^{Fft2} (see below, **figure a**) or Fun30^{Fft3} (see below, **figure b**). The results showed that ORFs with high RNAPII occupancy overlap almost exclusively with those with high enrichment of Fun30^{Fft2} or Fun30^{Fft3}.

Specific comment: 3. Did the authors find that both Fft2 and Fft3 are recruited to same gene-sets or do they occupy different sets of genes?

Our Response: To address the reviewer's question, we compared the enrichments of Fun30^{Fft2} at ORFs with those of Fun30^{Fft3} at ORFs using scatterplot analysis (see below). In this analysis, we found a strong correlation between their localizations ($r = 0.85$). Thus, we can conclude that Fun30^{Fft2} and Fun30^{Fft3} are recruited largely to the same gene sets. However, we also found that Fun30^{Fft2} shows localizations outside ORFs in a distinct manner from Fun30^{Fft3}. We included a new figure showing the similar and distinct localizations of Fun30^{Fft2} and Fun30^{Fft3} (**Supplementary Fig. 2c**) and discussed this point in the revised manuscript.

Specific comment: 4. The authors show changes in Pol II occupancy in *fft3Δ* strains. Did they measure this also at 37 degree, as they did for the FACT mutants? This is particularly important considering recent genome-wide studies showing global changes in transcription when cells are exposed to heat-shock. Thus, it is possible that some of the differences observed by the authors may be attributed to the heat-shock treatment.

Our Response: The comparisons of RNAPII occupancy changes in *fft3Δ* cells with those in *spt16-1* cells shown in **Fig. 2** were made by using cell grown at 37 °C. Thus, the differences observed from these cells are not attributable to temperatures. Instead, we suspected that relatively weak alleles of FACT and Fun30 (*spt16-1* and *fft3Δ*) used in this analysis might have caused the incomplete overlap between RNAPII occupancy changes of each mutant. To circumvent this, we performed ChIP-seq

analysis of RNAPII again using *spt16-18* cells and *fft2Δ fft3Δ* cells which are more defective in the function of FACT and Fun30, respectively. Both cells were grown at 37 °C for 2 hours after the temperature shift. As shown in the figure below (included in the revised manuscript as **Supplementary Fig. 2d**), we found that RNAPII occupancy changes of *fft2Δ fft3Δ* cells at ORFs positively correlate with those of *spt16-18* cells at ORFs. This indicates that Fun30 paralogs regulate RNAPII occupancy in a manner similar to FACT largely among the protein coding genes. Inspired by this result, we also compared the enrichment of FACT (Spt16-GFP) and Fun30 paralogs (Fft2-5xFLAG and Fft3-5xFLAG) at ORFs, which revealed positive correlations between their localizations to ORFs (**Supplementary Fig. 2e** in the revised manuscript). We discussed these results in the main text together with two new figures (**Supplementary Fig. 2d,e**).

Figure 2.

Specific comment: In this figure, the authors investigated Pol II occupancy in the *fft3Δ* and *spt16-1* single and *fft3Δ/spt16-1* double mutant to evaluate possible cooperation between remodelers and histone chaperone. Although, growth assay performed at the permissive temperature showed a synthetic growth phenotype, the effect of the double mutant on Pol II occupancy was very similar to that of the *spt16-1* single mutant. Thus, it appears that FACT may play a dominant role in regulating transcription, and the impact of lacking Fun30 is masked by the severe phenotype displayed by the FACT mutant. Their analysis identified a significant number of genes displaying Pol II occupancy defect $< -0.5 \log_2$ both in *fft3Δ* and FACT mutant. However, their analysis also suggests that there are

genes, which elicit greater reduction on deleting *Ftt3* than impairing *Spt16*. Have the authors analyzed whether the genes showing reduced Pol II occupancy in *fft3Δ* are indeed *Ftt3* targets. In other words, what is the level of enrichment of *Ftt3* at these genes in WT cells?

Our Response: To address the reviewer’s question, we performed Venn diagram analysis of the overlap between ORFs with high Fun30^{Ftt3} enrichment and those whose RNAPII occupancy is regulated by both Fun30^{Ftt3} and *Spt16* or by Fun30^{Ftt3} alone. We found that ORFs at which Fun30^{Ftt3} was highly enriched significantly overlapped with ORFs whose RNAPII occupancies were reduced in both *fft3Δ* cells and *spt16-1* cells, but not with those whose RNAPII occupancies were reduced only in *fft3Δ* cells (see below, **figure a**; included as **Fig. 2d** in the revised manuscript). Furthermore, we found that the levels of decreased RNAPII occupancy in *fft3Δ* cells at the ORFs whose RNAPII occupancies were reduced only in *fft3Δ* cells did not obey the trend of transcription-dependent decrease in RNAPII occupancy observed in *fft3Δ* cells (see below, **figure b**; included as **Supplementary Fig. 3d** in the revised manuscript). These findings suggest that direct regulation of RNAPII occupancy by Fun30^{Ftt3} during transcription is mediated in a FACT-dependent manner. We discussed these analyses in the main text of the revised manuscript.

Figure 4.

Specific comment: 1. In this figure, the authors have compared the changes in H3 occupancy in the *Fun30* mutant with the histone exchange over ORFs and found these to be positively correlated. They further suggest that the reduction in Pol II occupancy in *fun30* mutant is caused by defective nucleosome occupancy. This is primarily based on analysis carried out on genes showing higher histone exchange. It is difficult to assess the degree of changes in H3 occupancy or Pol II occupancy that occurs in mutant. The authors should explicitly mention how they choose these genes, and how much defect in H3 occupancy was observed at these genes in the *fit3Δ* mutant.

Our Response: As described earlier, we introduced a mathematical model to estimate the levels of nucleosome disassembly by $\text{Fun30}^{\text{Fit3}}$ at ORFs based on the available experimental data (see **Fig. 4c** and **Mathematical model in Methods** in the revised manuscript). We performed scatterplot analyses according to the estimation and found that the levels of nucleosome disassembly by $\text{Fun30}^{\text{Fit3}}$ at ORFs correlate with RNAPII occupancy changes in *fft3Δ* cells as well as RNAPII occupancies among the whole protein-coding genes (see **Fig. 4d,e** in the revised manuscript). Thus, selection of a specific subset of genes was no longer required to show the correlation between the function of $\text{Fun30}^{\text{Fit3}}$ in nucleosome disassembly at ORFs and the function of $\text{Fun30}^{\text{Fit3}}$ in regulation of RNAPII occupancy. We therefore removed some of the figures related to selection of the specific subset of genes (**Fig. 4a,c,d,e,f,g** in the original manuscript) and replaced them with new figures (**Fig. 4c,d,e** in the revised manuscript). We accordingly modified the main text.

Specific comment: 2. They should also specify whether the genes undergoing histone exchange were among those which showed *Fun30* recruitment.

Our Response: To address this point, we had to compare the enrichment of $\text{Fun30}^{\text{Fit3}}$ at ORFs with the rate of histone exchange. Since the rate of histone exchange was measured at 37°C using a temperature-sensitive cell cycle mutant (*cdc25-22*), the enrichment of $\text{Fun30}^{\text{Fit3}}$ at ORFs should also

be measured at 37°C to conduct a proper comparison. We have tried ChIP-seq for Fun30^{Fit3} in cells grown at 37°C during the course of the revision, but failed to obtain a good ChIP-seq signal probably because Fun30^{Fit3} binding to chromatin is too dynamic at a high temperature so that the cross linking of Fun30^{Fit3} to DNA may not be very efficient to give a good ChIP-seq signal. Instead, we tried to compare the enrichments of Fun30^{Fit3} at ORFs measured at 30°C with the rates of histone exchange measured at 37°C and found a positive correlation ($r = 0.45$; see below). Despite the correlation, we decided not to include this result because the comparison was made in cells grown at different temperatures and also because our estimation of Fun30^{Fit3}-mediated nucleosome disassembly improved the same point in a different way.

Specific comment: 3. (Page 13, line 266) The authors mention that “the reduced RNAPII occupancy defect.... among those that undergo rapid histone turnover”. And later (page 14, lines 287-291) they suggest that Fun30-triggered a nucleosome loss may have been underrepresented at many genes due to efficient reassembly by the FACT complex. The latter statement implies that there is active histone disassembly and reassembly, and thus what authors may be looking at in their histone exchange data is the integration of newly synthesized histones rather than the ones which are removed during transcription. This will also fit with previous observations that highest loss of histones is generally seen at strongly transcribed ORFs.

Our Response: With regard to the correlation between the rate of histone exchange and histone H3 occupancy changes in *fft3Δ* cells, the correlation is limited ($r = 0.41$) because the rate of histone exchange is measured by the rate of new histone incorporation, thereby linked to nucleosome loss only in an indirect manner. It is possible that nucleosome loss at ORFs does not always cause incorporation of new histones. We discussed this point in the main text of the revised manuscript. To more comprehensively investigate the role of Fun30^{FFt3} in the regulation of nucleosome occupancy at ORFs, in the revised manuscript we estimated the levels of Fun30^{FFt3}-mediated nucleosome disassembly using a mathematical model which integrates nucleosome disassembly, loss and reassembly during transcription altogether. Scatterplot analyses revealed that the levels of nucleosome disassembly by Fun30^{FFt3} at ORFs correlate with RNAPII occupancy changes in *fft3Δ* cells as well as RNAPII occupancies among the whole protein-coding genes (see **Fig. 4d,e** in the revised manuscript). In addition, our estimation of the levels of Fun30^{FFt3}-mediated nucleosome disassembly at ORFs (84% and 54% for RNAPII-CTD_{S2P} and RNAPII-CTD_{un}) significantly improved the conclusion that Fun30^{FFt3} plays a major role in RNAPII-mediated nucleosome disassembly at ORFs. According to these new analyses, we modified the main text of the revised manuscript.

Figure 5.

Specific comment: *Suppression of the cold-sensitive phenotype by deletion of two of the three H3/H4 copies strongly supports authors claim that Fun30 is involved in nucleosome disassembly. However, it is puzzling to see that slightly higher histone occupancy is observed in coding regions of genes included in Fig 5b. One would expect the ratio (mutant/WT) to be lower, considering that both histone levels and histone occupancies are reduced in H3/H4ΔΔ strain (suppl. Fig 5). Could authors comment on this observation? Do the authors see significant changes in histone occupancy in H3/H4ΔΔ vs WT strain?*

Our Response: Thank you for making us aware of this issue. Owing to the reviewer's comment, we noticed that there was a mistake in the processing of this particular ChIP-seq experiment, more

specifically during bigwig file generation using Deeptools. We corrected this and found that histone H3 occupancies were clearly reduced at ORFs in the *H3/H4ΔΔ* mutant. We accordingly modified **Fig. 5b** in the revised manuscript.

Again, we thank the reviewers for their time and effort spent reviewing our manuscript. The comments were astute and, in addressing them, we have greatly improved our insight into the function of chromatin remodeler Fun30.

P.S. The following is a list of changes made to the revised manuscript:

- 1) In order to respond to the general comment of Reviewer #2, we removed the non-essential statements about ‘general role of nucleosome in DNA transactions’ and ‘preventing transcription initiation at promoters’ in **Introduction**. In consequence, this editing removed the sentences commented by reviewer #2 (specific comment 2, 3).
- 2) In order to respond to Reviewer #2’s specific comments 4, 5, 6, 11, 29, references were added properly according to reviewer’s suggests. In addition, as Reviewer #2’s specific comment 22, incomplete reference was also revised to be complete one.
- 3) In order to respond to Reviewer #2’s specific comment 7, additional spotting assays were performed in various stress conditions to validate the choice of *fft2Δfft3Δ* strain as the mutant which is functionally defective to chromatin remodeler subfamily Fun30 in fission yeast (see revised **Supplementary Fig. 1**).
- 4) In order to respond to Reviewer #2’s specific comment 1, the sentence “We considered the absolute values of correlation coefficients greater than 0.4 ($r \geq 0.4$) as a meaningful correlation” was added at the description of the first scatterplot analysis results (revised **Fig. 1b**) in main text.
- 5) In order to respond to Reviewer #2’s specific comment 8 and Reviewer #3’s specific comment 1 for Fig.1, the clustering method was described in the legend of **Fig. 1a,c**, **Fig. 4a** and **Supplementary Fig. 2a,b** and the numbers of genes in each groups were described within the figures.

- 6) In order to respond to Reviewer #2's specific comment 12 and Reviewer #3's specific comment 3 for Fig. 1, genome-wide view analysis was performed to introduce example of different and similar binding pattern between Fun30^{Fft2} and Fun30^{Fft3} (**Supplementary Fig. 2c**). In conclusion, this result suggests that the enrichment difference between Fun30^{Fft2} and Fun30^{Fft3} is determined by their functional difference, not expression level. See main text of revised manuscript for more detail.
- 7) In order to respond to Reviewer #2's specific comment 14 and 16 and Reviewer #3's specific comment 4 for Fig. 1, scatterplot analyses were performed to show the correlation of RNAPII occupancy change in *fft2Δ fft3Δ* cells and in *Spt16-18* cells (**Supplementary Fig. 2d**), and the correlation of enrichment level of Fun30^{Fft2}, Fun30^{Fft3} and Spt16 at gene body regions of total protein coding genes (**Supplementary Fig. 2e**).
- 8) The word "high" was added into right position (at the legend of **Fig. 1a**) according to Reviewer #2's specific comment 15.
- 9) In order to respond to Reviewer #3's general comment, we performed spotting assay to illustrate that *fft3Δ* shows much stronger genetic interaction with *spt16-1* than *spt6-1* and *paf1Δ* (**Fig. 2a**). This result suggests that among the transcription elongation-related factors Spt16, Spt6 and PAF complex, Spt16 has distinct functional interaction with Fun30^{Fft3}.
- 10) In order to respond to Reviewer #3's specific comment for Fig. 2, we performed Venn diagram analyses to show that the genes with high Fun30^{Fft3} enrichment were dominantly overlapped by the genes of which RNAPII occupancy was regulated by both Fun30^{Fft3} and Spt16 than the genes of which RNAPII occupancy was regulated by Fun30^{Fft3} only (**Fig. 2d**). In addition, we also performed scatterplot analyses to illustrate that the genes of which RNAPII occupancy was regulated by both Fun30^{Fft3} and Spt16 showed clearer transcription-dependent RNAPII occupancy decrease in *fft3Δ* than the genes of which RNAPII occupancy was regulated by Fun30^{Fft3} only (**Supplementary Fig. 3d**).
- 11) Revise the improper sentence as "Consistent with the physical interaction data, we observed that the occupancy of Fun30^{Fft3} at transcribing regions strongly correlated with the occupancies of Set2 and the levels of Set2-mediated histone H3K36 trimethylation (H3K36me³)" according to Reviewer #2's specific comment 21.

- 12) Since previous address to explain the reason of mismatch between nucleosome loss by Fun30^{Fft3} and RNAPII occupancy (see revised **Fig. 4a**) was not enough to respond to the comments of all three reviewers (Reviewer #1's specific comments for Fig. 4a, g, Reviewer #2's specific comments 24 and 26, and Reviewer #3's specific comments 1~4 for Fig4), we revise everything of the main text for nucleosome disassembly function of Fun30^{Fft3} and **Fig. 4, Supplementary Fig. 4** using simple mathematical approach described in **Mathematical model** of **Methods**. In addition, during this editing, incorrect sentences within this section were removed to respond to the Reviwer #2's specific comments 19 and 20. Please read the revised manuscript to understand the detail.
- 13) In order to respond to Reviewer #2's specific comment 27 and Reviewer #3's specific comment for **Fig. 5**, we corrected **Fig. 5b** since there was minor error in processing raw data for this analysis.
- 14) In order to respond to Reviewer #2's specific comment 28, we added brief introduction to Pst1 and Cph1 of the Clr6 complex II in the main text as followed: "Subunits of Clr6 complex II are non-essential except the catalytic subunit Clr6. A subunit of Clr6 complex II, Cph1 is a fission yeast homolog of budding yeast Rco1, a PHD zinc-finger domain containing protein of Rpd3S complex. ... Another subunit Pst2 is a fission yeast homolog of Sin3 proteins which are conserved from budding yeast to mammals."
- 15) In order to respond to Reviewer #2's specific comment 9, revise the description to RNAPII-CTDun and RNAPII-CTDS2P antibodies to be more explicit as followed: "Antibodies against α -RNA polymerase II CTD repeat YSPTSPS without phosphorylation (8WG16; 920102, BioLegend; previously MMS-126R, Covance), the α -RNA polymerase II CTD repeat YSPTSPS with phosphorylation at Ser2 (phospho S2; ab5095, Abcam) ..."
- 16) In order to respond to Reviewer #2's specific comment 30, we noticed that no spike-in control was used in this study in the **ChIP-seq analysis** section of **Methods** part as followed: "Spike-in controls were not included in our ChIP-seq samples. Thus, it should be noted that the ChIP-seq signals in our analysis represent a relative rather than an absolute measurement of enrichment."

REVIEWERS' COMMENTS:

Reviewer #1 (Remarks to the Author):

I am satisfied with the changes that have been made to the manuscript.

Reviewer #2 (Remarks to the Author)

The authors have overall done a nice job of addressing the comments. A few remaining points are listed below. My comments are below the authors' responses.

Specific comment: 1. A lot of the conclusions in this work rest on the evaluation of correlation coefficients. In some cases, it seems that the authors are making a strong conclusion based on r values that may not merit it – the most striking example of this is in Fig. 4b. Compare Fig. 4b ($r=0.41$) and Supp. Fig. 4a ($r=0.25$). The first is discussed as meaningful and the second as not meaningful. Also in many cases, we are asked to evaluate the r values for comparisons between chip-seq of different proteins when we haven't been given the r values for the replicate experiments. We need those values as well. A clearer discussion of what the authors consider to be meaningful r values, as well as more information about duplicates should be provided throughout the manuscript.

Our Response: We interpreted the correlation coefficients greater than 0.4 ($r \geq 0.4$) to indicate a positive correlation according to a widely-used interpretation of the correlation coefficients in statistics. Generally, r value 0.00 ~ 0.19 is interpreted as very weak, 0.20 ~ 0.39 to be weak, 0.40 ~ 0.59 to be moderate, 0.60 ~ 0.79 to be strong, and 0.80 ~ 1.00 to be very strong (Evans, James D. Straightforward statistics for the behavioral sciences. Brooks/Cole, 1996.). Based on this interpretation, the correlation coefficients less than 0.4 ($r < 0.4$) are weak or very weak, so that we considered them not sufficient to indicate a positive correlation. We explicitly described this in the main text of revised manuscript: "We considered the absolute values of correlation coefficients greater than 0.4 ($r \geq 0.4$) as a meaningful correlation". In addition to this, when we merged ChIP-seq data of biological duplicates we validated that they are sufficiently correlative (the minimum r between duplicates: 0.92). Below, we added a table showing the correlation coefficients determined between ChIP-seq data of biological duplicates.

Reviewer comment: I don't think that most readers will find $r=0.41$ to be too convincing, so I recommend citing the Evans reference when you state your conclusion on line 148.

Specific comment: 24. Fig. 4 and page 12 – There is a lot of information presented in terms of correlation coefficients and ECDF plots. The authors should provide some specific numbers: for example, in *fft3* mutants, how many genes have increased H3 levels and how many genes do not, and also in the case of comparing histone exchange rate to histone levels in the *fft3* mutant. With that correlation coefficient, and based on the scatterplot, there are many genes that have the opposite effect. This should be discussed.

Our Response: As described earlier, we introduced a mathematical model to estimate the levels of nucleosome disassembly by Fun30Fft3 at ORFs based on the available experimental data (see Fig. 4c and Mathematical model in Methods in the revised manuscript). We performed scatterplot analyses according to the estimation and found that the levels of nucleosome disassembly by Fun30Fft3 at ORFs correlate with RNAPII occupancy changes in *fft3Δ* cells as well as RNAPII occupancies among the whole protein-coding genes (see Fig. 4d,e in the revised manuscript). Thus, selection of a specific subset of genes was no longer required to show the correlation between the function of Fun30Fft3 in nucleosome disassembly at ORFs and the function of Fun30Fft3 in regulation of RNAPII occupancy. We therefore removed some of the figures related to selection of the specific subset of genes (Fig. 4a,c,d,e,f,g in the original manuscript) and replaced them with new figures (Fig. 4c,d,e in the revised manuscript). We accordingly modified the main text. With regard to the correlation between the rate of histone exchange and histone H3 occupancy changes in *fft3Δ* cells, the correlation is limited ($r = 0.41$) because the rate of histone exchange is measured by the rate of new histone incorporation, thereby linked to nucleosome loss only in an indirect manner. It is possible that nucleosome loss at ORFs does not always cause incorporation of new histones.

Reviewer response: This formula and its explanation were not clear to me. Where do the values for D, L, and R come from? What's the reference for the statement that *spt16-18* lacks nucleosome assembly function?

Specific comment: 30. Several recent sets of studies have provided evidence that spike-in controls are important when measuring the genome-wide effects of potentially global factors. For example, see Chen et al. (MCB 36, 662) and Loven et al. (Cell 151, 476). As those weren't included in these experiments, the authors should consider mentioning that as a caveat.

Our Response: According to the reviewer's suggestion, we mentioned about this in the ChIP-seq analysis section of Methods as follows. "Spike-in controls were not included in our ChIP-seq samples. Thus, it should be noted that the ChIP-seq signals in our analysis represent a relative rather than an absolute measurement of enrichment."

Reviewer response: I think the statement should be further revised. A suggestion: "Spike-in controls were not included in our ChIP-seq samples. Thus, it should be noted that the relative ChIP-seq signals in our analysis may not be sensitive to a global change."

Reviewer #3 (Remarks to the Author):

The revised manuscript is substantially improved after incorporating new data and additional interpretations. I am satisfied with their responses to the concerns raised in previous version.

REVIEWERS' COMMENTS:

Reviewer #1 (Remarks to the Author):

I am satisfied with the changes that have been made to the manuscript.

Our response: Thank you.

Reviewer #2 (Remarks to the Author)

The authors have overall done a nice job of addressing the comments. A few remaining points are listed below. My comments are below the authors' responses.

Specific comment: 1. *A lot of the conclusions in this work rest on the evaluation of correlation coefficients. In some cases, it seems that the authors are making a strong conclusion based on r values that may not merit it – the most striking example of this is in Fig. 4b. Compare Fig. 4b ($r=0.41$) and Supp. Fig. 4a ($r=0.25$). The first is discussed as meaningful and the second as not meaningful. Also in many cases, we are asked to evaluate the r values for comparisons between chip-seq of different proteins when we haven't been given the r values for the replicate experiments. We need those values as well. A clearer discussion of what the authors consider to be meaningful r values, as well as more information about duplicates should be provided throughout the manuscript.*

Our Response: *We interpreted the correlation coefficients greater than 0.4 ($r \geq 0.4$) to indicate a positive correlation according to a widely-used interpretation of the correlation coefficients in statistics. Generally, r value 0.00 ~ 0.19 is interpreted as very weak, 0.20 ~ 0.39 to be weak, 0.40 ~ 0.59 to be moderate, 0.60 ~ 0.79 to be strong, and 0.80 ~ 1.00 to be very strong (Evans, James D. Straightforward statistics for the behavioral sciences. Brooks/Cole, 1996.). Based on this interpretation, the correlation coefficients less than 0.4 ($r < 0.4$) are weak or very weak, so that we considered them not sufficient to indicate a positive correlation. We explicitly described this in the main text of revised manuscript: “We considered the absolute values of correlation coefficients*

greater than 0.4 ($r \geq 0.4$) as a meaningful correlation". In addition to this, when we merged ChIP-seq data of biological duplicates we validated that they are sufficiently correlative (the minimum r between duplicates: 0.92). Below, we added a table showing the correlation coefficients determined between ChIP-seq data of biological duplicates.

Reviewer comment: I don't think that most readers will find $r=0.41$ to be too convincing, so I recommend citing the Evans reference when you state your conclusion on line 148.

Our response: Thank you for your suggestion. As you recommended, we cited the Evans reference (Evans, James D. Straightforward statistics for the behavioral sciences. Brooks/Cole, 1996) on that sentence.

Specific comment: 24. Fig. 4 and page 12 – There is a lot of information presented in terms of correlation coefficients and ECDF plots. The authors should provide some specific numbers: for example, in *fft3* mutants, how many genes have increased H3 levels and how many genes do not, and also in the case of comparing histone exchange rate to histone levels in the *fft3* mutant. With that correlation coefficient, and based on the scatterplot, there are many genes that have the opposite effect. This should be discussed.

Our Response: As described earlier, we introduced a mathematical model to estimate the levels of nucleosome disassembly by *Fun30^{fft3}* at ORFs based on the available experimental data (see Fig. 4c and Mathematical model in Methods in the revised manuscript). We performed scatterplot analyses according to the estimation and found that the levels of nucleosome disassembly by *Fun30^{fft3}* at ORFs correlate with RNAPII occupancy changes in *fft3Δ* cells as well as RNAPII occupancies among the whole protein-coding genes (see Fig. 4d,e in the revised manuscript). Thus, selection of a specific subset of genes was no longer required to show the correlation between the function of *Fun30^{fft3}* in nucleosome disassembly at ORFs and the function of *Fun30^{fft3}* in regulation of RNAPII occupancy. We therefore removed some of the figures related to selection of the specific subset of genes (Fig. 4a,c,d,e,f,g in the original manuscript) and replaced them with new figures (Fig. 4c,d,e in the revised

manuscript). We accordingly modified the main text. With regard to the correlation between the rate of histone exchange and histone H3 occupancy changes in *fft3Δ* cells, the correlation is limited ($r = 0.41$) because the rate of histone exchange is measured by the rate of new histone incorporation, thereby linked to nucleosome loss only in an indirect manner. It is possible that nucleosome loss at ORFs does not always cause incorporation of new histones.

Reviewer response: This formula and its explanation were not clear to me. Where do the values for D , L , and R come from? What's the reference for the statement that *spt16-18* lacks nucleosome assembly function?

Our response: The formula and its explanation were newly introduced in the revised manuscript to strengthen Figure 4. We explicitly described how we define the parameters D , L and R in the main text and in **Methods**. We also described how we derived the formula in **Methods**. We hope that you could find our explanations for the formula, especially “**Mathematical model**” in the **Methods** section, helpful. The evidence demonstrating defective nucleosome reassembly during RNAPII elongation in *spt16-18* cells is provided by our own data (**Supplementary Figure 4b**), but we also added a new reference which made the same conclusion in a previous study (Choi *et al.*, 2012).

Specific comment: 30. Several recent sets of studies have provided evidence that spike-in controls are important when measuring the genome-wide effects of potentially global factors. For example, see Chen *et al.* (MCB 36, 662) and Loven *et al.* (Cell 151, 476). As those weren't included in these experiments, the authors should consider mentioning that as a caveat.

Our Response: According to the reviewer's suggestion, we mentioned about this in the ChIP-seq analysis section of **Methods** as follows. “Spike-in controls were not included in our ChIP-seq samples. Thus, it should be noted that the ChIP-seq signals in our analysis represent a relative rather than an absolute measurement of enrichment.”

Reviewer response: *I think the statement should be further revised. A suggestion: “Spike-in controls were not included in our ChIP-seq samples. Thus, it should be noted that the relative ChIP-seq signals in our analysis may not be sensitive to a global change.”*

Our response: Thank you for your kind recommendation. We revised the statement as your suggestion.

Reviewer #3 (Remarks to the Author):

The revised manuscript is substantially improved after incorporating new data and additional interpretations. I am satisfied with their responses to the concerns raised in previous version.

Our response: Thank you.